# Photoactivated conductive MOF thin film arrays on micro-LEDs for chemiresistive gas sensing

Kichul Lee [1,7], Young-Moo Jo [2,3,7] ✉, Myung Sung Sohn[2,7], Mingyu Jeon[4], Cheolmin Kim[1], Osman Gul [1], Seon Ju Park[2], Ki Beom Kim[2], Ki Soo Chang[5,6], Chan Bae Jeong [5], Jihan Kim [4], Yun Chan Kang[2] ✉ & Inkyu Park [1] ✉

Electrically conductive metal-organic frameworks (cMOFs) are emerging as promising chemiresistors due to their diverse compositions, chemical properties, porosity, and room-temperature conductivity, enabling the design of energy-efficient devices. However, limited activation in this regime hinders sensitivity and reversibility. In this study, cMOF thin films are integrated onto a micro-LED (μLED) platform using a layer-by-layer method, enabling photo-activated gas sensing even at room-temperature. The systematic coating allows for precise tailoring of films (e.g., thickness and overlayer structures) based on the adsorption properties of each analyte (ethanol, trimethylamine, ammonia, nitrogen dioxide). The selected arrays are optimized by varying the wavelengths and intensities of μLED, enabling sensitive and reversible sensing through additional charge generation, while consuming ultra-low power (587 μW). Additionally, a deep learning algorithm achieves rapid gas recognition within tens of seconds, with 99.8% classification accuracy in concentration prediction. This work demonstrates the feasibility of the cMOF−μLED integrated sensor platform, paving the way for next-generation gas-sensing technologies

Electrically conductive metal-organic frameworks (cMOFs) have significantly broadened their applications in fields requiring conductivity, including electrocatalysis, chemical sensors, and energy storage devices[1–6]. Notably, they are highly effective for chemiresistive gas sensing due to advantages such as high gas accessibility through abundant pores and enhanced reactions at catalytic sites (metal clusters and ligand functional groups) distributed across a large surface area[7–10]. Additionally, their functional tunability, which can be easily modified by substituting metals or ligands[11,12], provides the opportunity to develop tailored sensing materials for various target gases.

Integrating cMOFs into gas sensor arrays allows for detecting multiple gases simultaneously, gathering ambient chemical information for environmental monitoring, industrial safety, and medical diagnostics[13,14].

Light activation of chemiresistors has been gaining attention owing to its ability to enhance sorbate-sorbent interactions and accelerate recovery reactions of adsorptive gases[15–17]. Notably, the inherent optical properties of cMOFs, arising from various transition states (e.g., metal-to-ligand charge transfer or π-π* transition), can be sensitively activated by exposure to the relevant spectrum of light[18,19].

[1]Department of Mechanical Engineering, Korea Advanced Institute of Science and Technology (KAIST), Daejeon, Republic of Korea. [2]Department of Materials Science and Engineering, Korea University, Seoul, Republic of Korea. [3]School of Materials Science and Engineering, Kyungpook National University, Daegu, Republic of Korea. [4]Department of Chemical and Biomolecular Engineering, Korea Advanced Institute of Science and Technology (KAIST), Daejeon, Republic of Korea. [5]Center for Scientific Instrumentation, Korea Basic Science Institute, Daejeon, Republic of Korea. [6]School of Electrical and Electronic Engineering, Chung-Ang University, Seoul, Republic of Korea. [7]These authors contributed equally: Kichul Lee, Young-Moo Jo, Myung Sung Sohn. ✉e-mail: jym754@knu.ac.kr; yckang@korea.ac.kr; inkyu@kaist.ac.kr

For example, the $Cu_3HHTP_2$ ($H_6HHTP$ = 2,3,6,7,10,11-hexahydroxytriphenylene) achieves rapid $NO_2$ recovery by generating charge carriers through the π–π* transition states, excited by blue light[20]. While the development of light-activated MOF chemiresistor arrays is still in its early stages, studies on luminescent or photocatalytic properties have revealed various relationships between the band gap of MOFs and light[21,22]. These support the rationality of charge transfer induced by light. It is worth noting that applying light activation on cMOFs has rarely been reported. Therefore, the vast combinations of emerging cMOFs and diverse light sources hold great promise for creating high-performance sensor arrays. Particularly, recent micro-sized light-emitting diodes (μLED) based sensor research has demonstrated the potential to drastically reduce the size and power consumption of devices, as shown for instance with monolithic multi-μLED platforms (μLPs) integrated with Au and Ag nanoparticles coated $In_2O_3$ sensor arrays[23–26]. By targeting the localized area necessary for photoactivation with μLEDs (width <100 μm) only, and placing the sensing material directly above the light source (distance: 1 μm), extremely high light energy transfer efficiency is achieved, enabling ultra-low power gas sensing at a level of 0.1 mW. By leveraging the synergistic effects of μLED technology and light-activated cMOF chemiresistors, a promising opportunity arises for the development of a highly accurate electronic nose (e-nose) system.

Herein, we developed an ultra-low power μLED embedded gas sensor array fabricated with an $M_3HHTP_2$ (M = Co, Ni, Cu). To demonstrate the versatility of this cMOF-μLED integration system, optimization was conducted in two modulation parts: chemiresistor films and light sources. As depicted in Fig. 1a, 2D $M_3HHTP_2$ with hexagonal pores of approximately 2 nm was uniformly coated as thin films using the layer-by-layer (LBL) method. This method involves sequentially exposing the substrate to metal and ligand solutions, allowing for control over the configuration and composition of sensing films by adjusting the number and sequence of MOF coatings[27,28]. We adopted

$Cu_3HHTP_2$ as a conducting layer, because it is well known for its stable chemiresistive properties, which can be finely tuned by adjusting the LBL coating conditions[29–31]. The density and thickness of $Cu_3HHTP_2$ thin films can be modified depending on the number of coating cycles, enabling precise adjustment of the total number of gas reaction sites and gas accessibility. Additionally, substituting Cu with Ni or Co enables the synthesis of $Ni_3HHTP_2$ and $Co_3HHTP_2$ on $Cu_3HHTP_2$ layers, both of which share similar hexagonal structures. Considering that the use of $Ni_3HHTP_2$ and $Co_3HHTP_2$ as thin conducting films is challenging because of their higher intrinsic resistance than that of $Cu_3HHTP_2$[6], we propose constructing them as overlayered film configurations using the LBL method. This approach serves as a strategic means to diversify gas-sensing characteristics, with the upper layer of the sensing film functioning as a catalytic layer to facilitate reactions with the target gases[15,32,33]. These catalytic properties were proven using density functional theory (DFT) calculations.

In this study, we introduce an e-nose system capable of distinguishing different gases by integrating photoactivated cMOF sensor arrays with a deep learning algorithm. In addition to optimizing the conditions of the cMOF sensing films, employing various μLED light sources with different wavelengths and intensities broadens the diversity of sensing signals (Fig. 1b). Light sources that emit photons with energies close to the absorption states of $Cu_3HHTP_2$, such as blue and ultraviolet (UV) ($\lambda_{peak}$ = 455 and 395 nm, respectively), are effective in generating photo-induced electron–hole pairs. Room-temperature gas sensors typically operate through direct redox reactions between the target gas and the chemiresistor, with additional charge generation potentially facilitating the gas reaction[20]. In our pursuit of developing the most discriminative sensor array combination, we meticulously considered a wide range of variables to simultaneously optimize the cMOF sensing films and the light source conditions (Fig. 1c). This optimization process considers factors such as resistance, sensitivity, selectivity, and reversibility, with a primary focus on achieving a

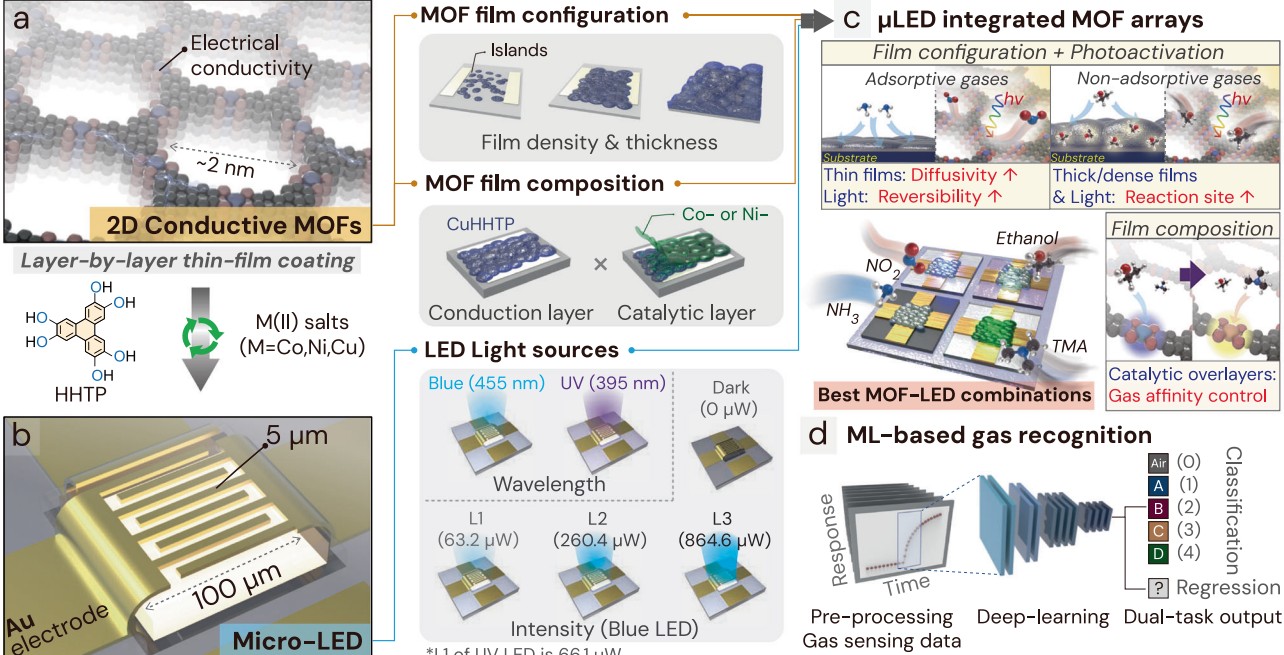

**Fig. 1 | Schematic overview of this study, illustrating the use of a 2D conductive metal-organic frameworks (cMOF) layer atop an ultra-low power micro light-emitting diode (μLED) gas sensor, which is employed as both a gas sensor and an e-nose. a** An illustration of the optimized 2D cMOF films for gas sensing, showing variations in film configuration (density and thickness) and composition (conduction layers and catalytic overlayers). **b** An illustration of optimization process of the light wavelength and intensity of μLED to maximize the gas-sensing capability of the cMOFs (width and spacing of electrodes: 5 μm). **c** An array of cMOFs composed of different sensors (varying μLED types, light intensities, and cMOF types). **d** Light-activated cMOF chemiresistive gas sensor array integrated with a deep learning-enabled e-nose system.

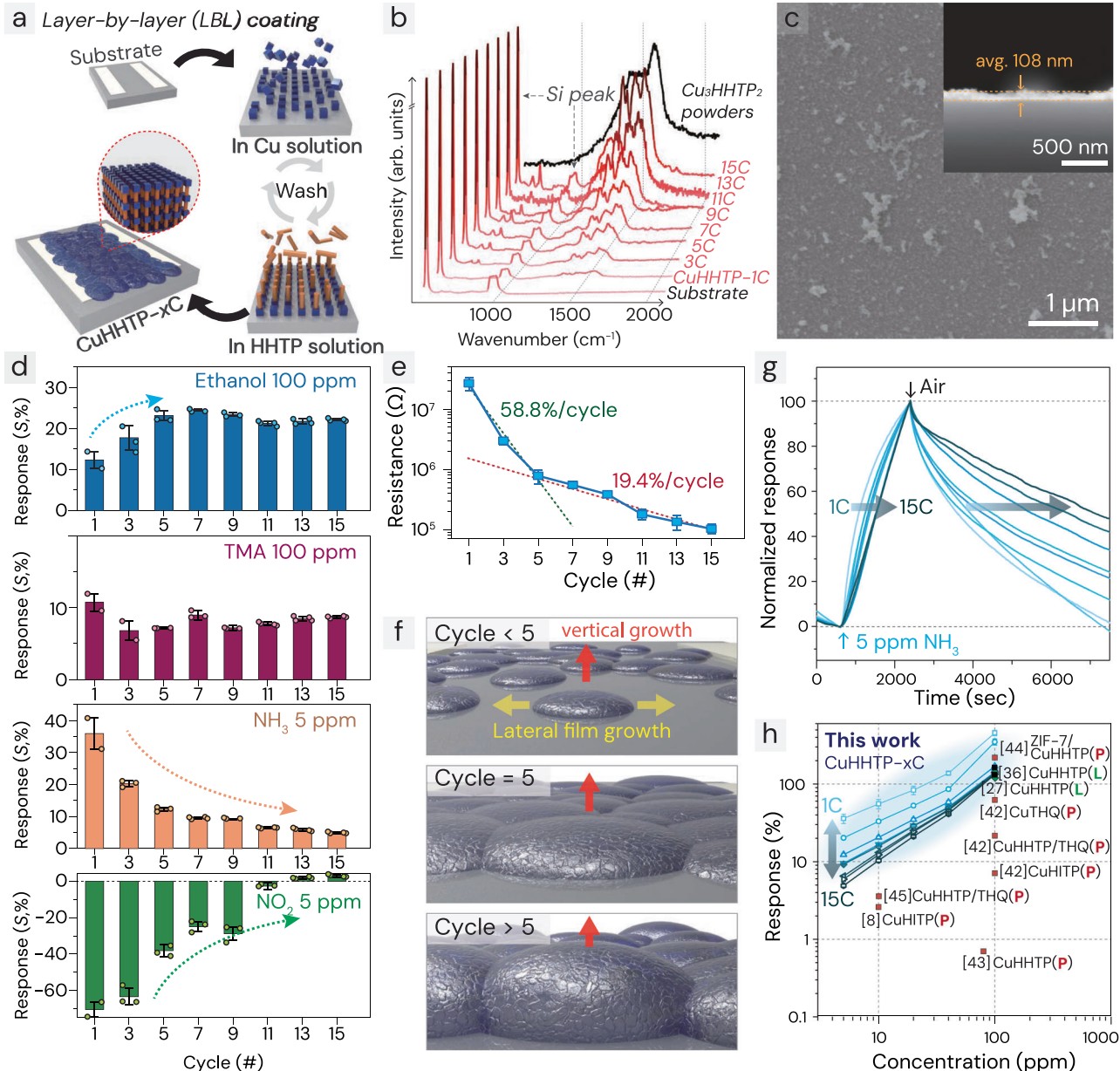

**Fig. 2 | Layer-by-layer (LBL) synthesis process, characterization, and gas testing results of Cu₃HHTP₂.** **a** Schematic of $Cu_3HHTP_2$ synthesis by repeating the LBL process of immersing in Cu and HHTP solutions (orange stick: ligand, navy cube: metal ion). **b** Raman spectroscopy results of $Cu_3HHTP_2$ powders and CuHHTP-xC films at different cycles ($x = 1, 3, 5, 7, 9, 11, 13,$ and 15). **c** Top and cross-sectional view SEM images of CuHHTP-5C. The thickness of CuHHTP-5C represents the mean value from 10 independent measurements. **d** Comparison of the response of CuHHTP-xC sensors to EtOH, TMA, $NH_3$, and $NO_2$. **e** Changes in base resistance with varying LBL cycles of CuHHTP-xC sensors, and **f** predicted scheme of film growth (in-plane to out-of-plane) with increasing LBL cycles. **g** Normalized response transients of CuHHTP-xC sensors to 5 ppm $NH_3$. **h** Comparison of CuHHTP-xC sensors in this study with previous cMOF-based gas sensors research (L LBL-coated films, P powders). **d, e, g, h** All results represent the average values from different sensors ($n = 2$ to 4). All error bars in the figures represent the standard deviation. Source data are provided as a Source data file.

discriminative gas sensor array by managing three key gas behaviors: (i) gas diffusivity, (ii) gas affinity, and (iii) reversibility. Through this approach, we present a strategy that distinguishes gases not only by differences in response values but also by time-dependent signal transients. To this end, we employed convolutional neural network (CNN), which is specialized in pattern recognition, to leverage real-time transient response signals and significantly enhance the gas-discrimination capabilities of the array (Fig. 1d). Harnessing the emergent potential of cMOFs and the ultra-compact, ultra-low power capabilities of μLEDs, this study lays the groundwork for e-nose technology, combining cMOF chemiresistor arrays with deep learning to advance gas detection.

## Results

### Synthesis of Cu₃HHTP₂ sensing films using LBL methods

Figure 2a depicts the LBL coating process for the $Cu_3HHTP_2$ sensing film. Before integrating the cMOF onto the μLED, the LBL synthesis conditions were optimized using Au-interdigitated electrodes (IDEs) on bare Si substrates without light-emitting components (Supplementary Fig. 1). The substrates were sequentially immersed in ethanol solutions containing Cu acetate (1.0 mM) and $H_6HHTP$ (0.2 mM) for 1 and 2 min, respectively. To achieve uniform and smooth film surfaces, the substrates were immediately rinsed with clean ethanol following each exposure to remove any residual salts and prevent unwanted growth. $Cu_3HHTP_2$ films were applied via the LBL method across

various cycle numbers ($x = 1$, 3, 5, 7, 9, 11, 13, and 15), denoted as CuHHTP-$x$C, enabling an investigation of their thickness-dependent characteristics. The phase of CuHHTP-$x$C films was confirmed by comparing with crystalline $Cu_3HHTP_2$ powders via Raman spectroscopy (Fig. 2b and Supplementary Fig. 2). The peak positions of $Cu_3HHTP_2$ films were matched with that of powders, and their intensity was gradually increased in proportional to the cycle numbers. The presence of Cu, C, and O was confirmed by X-ray photoelectron spectroscopy (XPS), and Cu was found to exist in multiple valence states as verified by high-resolution XPS (Supplementary Fig. 3). Scanning electron microscope (SEM) images revealed that the thickness of $Cu_3HHTP_2$ films were increased with the number of coating cycles (Supplementary Fig. 4). For example, the thickness of CuHHTP-5C is shown in Fig. 2C, measuring 108 nm.

### Thickness-dependent gas-sensing characteristics of CuHHTP-$x$C sensors

The gas-sensing characteristics of CuHHTP-$x$C sensors were examined using a homemade gas-sensing setup (Supplementary Fig. 1), with various gases including ethanol (EtOH; $C_2H_5OH$), trimethylamine (TMA; $C_3H_9N$), ammonia ($NH_3$), and nitrogen dioxide ($NO_2$), which are representative airborne chemicals due to their environmental impact and health risks[34]. Additionally, the distinct adsorption properties of these gases—EtOH (neutral, non-adsorptive), TMA (basic, weakly adsorptive due to the steric hindrance of methyl groups), $NH_3$ (basic, adsorptive), $NO_2$ (acidic, adsorptive)—aid in understanding the correlation between cMOF film compositions and gas-sensing behaviors.

All sensors were stabilized in air, and the analytic gases were injected into the gas chamber for 30 min under dark condition at room temperature (Supplementary Fig. 5). The response ($S$) was calculated as $(R_g - R_a)/R_a$, where $R_g$ and $R_a$ are resistance in gas and air, respectively. The resistance variation of CuHHTP-$x$C sensors to EtOH, TMA, and $NH_3$ was positive, while it was negative to $NO_2$, which is consistent with the behavior of p-type chemiresistors[35]. Interestingly, it was found that the response values to each gas varied significantly depending on the number of cMOF coatings (Fig. 2d). First, the responses of CuHHTP-$x$C sensors toward 100 ppm EtOH gradually increased from 12.3% to 21.1% as the number of coatings increased from 1 to 5 cycles, and further coating did not increase the response but rather led to saturation. This result suggests that there were not enough reaction sites for ethanol in the initial stages of cMOF coating, which gradually increased as the number of coatings increased. As evidence of the insufficient amount of sensing material in the early cycles, the resistance ($R_a$) of the CuHHTP-$x$C sensors sharply decreased by 58.8% per cycle during cycles 1 to 5. However, after the 5th cycle, the rate of decrease significantly slowed to 19.4% per cycle (Fig. 2e). Although CuHHTP films are observed to form over a wide area from the very first cycle (Supplementary Fig. 6a), the internal connectivity of the film may still be insufficient at this stage, as the initial nuclei and primary particles have not fully grown or merged[36]. Consequently, during the early cycles, the steep drop in $R_a$ is likely due to the combined effects of thickness growth and lateral interconnection between the initially formed nuclei and/or primary particles. In contrast, at the higher coating cycles (>5 times), lateral film growth appears to reach saturation, and only vertical (thickness) growth continues—accounting for the slower rate of $R_a$ decrease (Fig. 2f). Furthermore, the gradual reduction in resistance could also be attributed to ongoing defect healing within the MOF throughout the entire LBL cycling process. For instance, Ma et al. demonstrated that defects of ZIF-8 membranes during the early LBL cycles are eliminated after 10 continuous cycles of LBL coating[37].

In contrast, the responses of CuHHTP-$x$C sensors to 5 ppm $NH_3$ and $NO_2$ exhibited reverse trends (Fig. 2d). The thinnest CuHHTP-1C sensors showed the highest responses of 36.0% for $NH_3$ and 70.4% for $NO_2$, but the responses continuously decreased with additional coatings. For examples, the responses of CuHHTP-5C sensors to $NH_3$ and $NO_2$ were 12.3% and 38.1%, respectively, and those of CuHHTP-15C sensors further decreased to 4.9% and 2.9%, respectively. These different trends compared to EtOH are presumably attributable to the adsorption properties of gas molecules that causes chemical filtration effect at the upper part of sensing films. Compared to EtOH, which is a neutral molecule, $NH_3$ and $NO_2$ are highly adsorptive due to their strong basic and acidic properties, respectively. Therefore, they are easily adsorbed at the surfaces of $Cu_3HHTP_2$ films, especially Cu sites[38,39], which amplifies the gas filtering effect, even at the initial stage of cMOF coatings. Interestingly, the TMA sensing properties remained almost constant regardless of the coating cycles. This is because TMA exhibits weaker adsorptivity than $NH_3$, due to the substitution of hydrogen atoms with three methyl groups. The methyl groups also attempt to competitively occupy the reaction sites, and their steric hindrance effect makes it difficult for the basic nitrogen atom to adsorb. It presumably results in intermediate response trends between non-adsorptive gases (such as EtOH) and highly adsorptive gases (such as $NH_3$).

From normalized response transients of CuHHTP-$x$C sensors exposed to 5 ppm of $NH_3$, we can identify both reaction and recovery speeds noticeably reduced as the thickness of $Cu_3HHTP_2$ film increased from 1 to 15 coating cycles (Fig. 2g). This result clearly indicates a gas filtering phenomenon due to strong gas adsorption. It should be noted that the recovery speed of $NO_2$ gas cannot be compared by thickness due to its irreversible behavior (Supplementary Fig. 7). This limitation highlights a significant challenge in achieving repeatable use for many room-temperature gas sensors[40,41]. Integrating cMOFs sensors with a light source, though, generates additional photogenerated charges that drive the reverse reaction, facilitating the desorption of adsorptive gases and thereby enhancing their reversibility to realize repeatable response and recovery[20].

The gas responses of $Cu_3HHTP_2$ sensors to $NH_3$ exhibited linear trends depending on the concentrations from 5 to 100 ppm (Fig. 2h). Furthermore, the responses of CuHHTP-1C and CuHHTP-3C to $NH_3$ were confirmed to be one of the highest response ($S = 457.1\%$ and 349.6%) among the reported room-temperature gas sensors that uses cMOFs (Fig. 2h)[8,27,36,42–45]. Therefore, the formation of thin films via LBL coating methods is the most advantageous approach to achieving superior sensing performance of cMOF chemiresistors compared to those prepared by powder deposition.

### Overlayer coating of $M_3HHTP_2$ (M = Ni, Co) on CuHHTP-5C films

$Ni_3HHTP_2$ and $Co_3HHTP_2$ films, which are analogs of $Cu_3HHTP_2$, were also fabricated using the same LBL procedures, because the different gas affinities of the Co and Ni metal sites can diversify the sensing properties of sensor arrays. However, they have limited application as standalone chemiresistive sensors due to their relatively low conductivity even after 15 LBL cycles. Instead, the conductive chemiresistive $Cu_3HHTP_2$ films were maintained, and $Ni_3HHTP_2$ and $Co_3HHTP_2$ were coated as overlayers, which is MOF-on-MOF epitaxial heterostructures[46], utilizing their catalytic properties. The $Cu_3HHTP_2$ films were coated for 5 cycles because this is the optimized number required to achieve high sensitivity for all analyte gases. To minimize the degradation of the base conductive $Cu_3HHTP_2$ films by heterometal ($Co^{2+}$ or $Ni^{2+}$) salt solutions, the LBL synthesis conditions for the overlayer coatings of $Ni_3HHTP_2$ and $Co_3HHTP_2$ were diluted tenfold (0.1 mM for metal solutions and 0.02 mM for ligand solutions). The gas-sensing performance was then assessed with additional $Ni_3HHTP_2$ and $Co_3HHTP_2$ coatings at 1, 3, 5, and 7 cycles, respectively, which is referred to as $MHHTP$-$y$C/CuHHTP-5C (M = Ni, Co; $y = 1$, 3, 5, 7) (Fig. 3a). As $Ni_3HHTP_2$ and $Co_3HHTP_2$ are coated, the increasing thickness compared to CuHHTP-5C indicates the formation of an overlayer (Supplementary Fig. 8). To confirm the overlayer structure, the MOF thin films were slightly tilted to simultaneously observe both the cross-

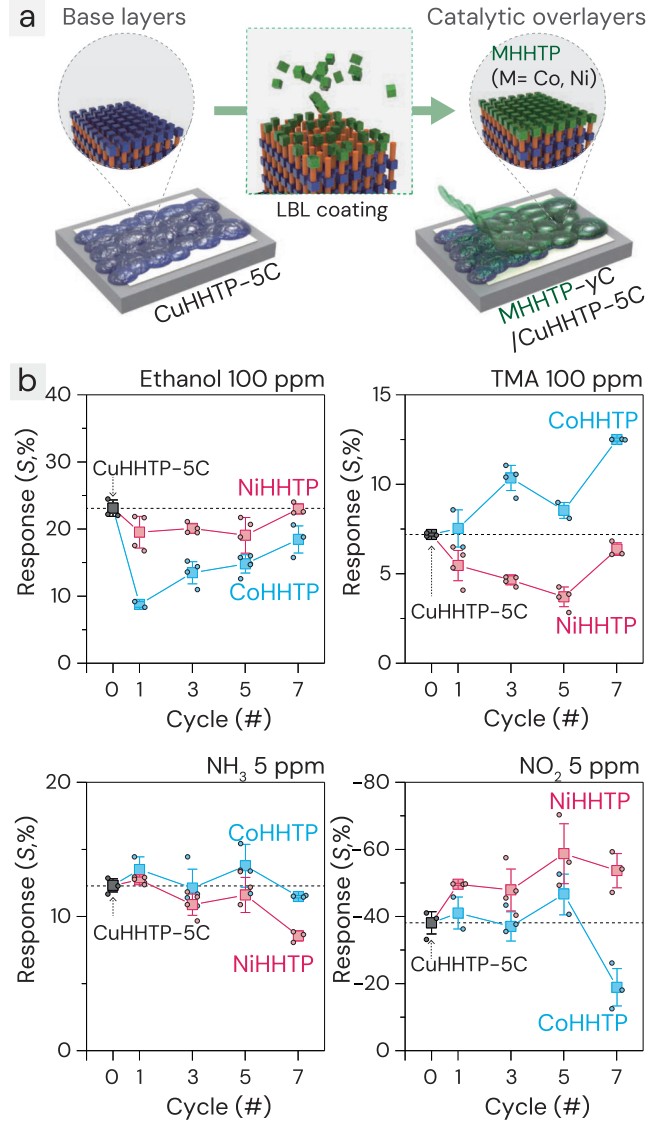

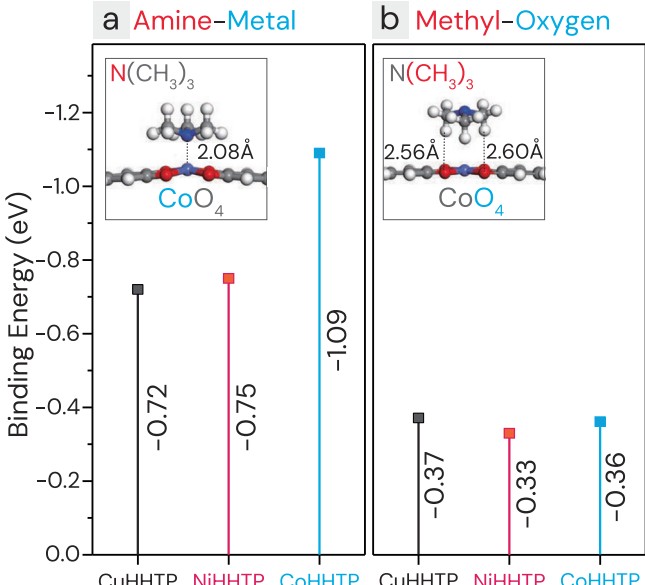

**Fig. 4 | DFT calculation of binding energy between a trimethylamine (TMA) gas and M₃HHTP₂ (M = Co, Ni, Cu) monolayer. a** Amine group-metal interaction. **b** Methyl group-oxygen interaction.

**Fig. 3 | Enhanced gas reactivity through overlayer cMOF coating on Cu₃HHTP₂ films. a** Process of creating MHHTP·$y$C/CuHHTP-5C (M = Ni, Co; $y$ = 1,3,5,7) by coating a catalytic overlayer on the Cu₃HHTP₂ conductive layer using the LBL method (orange stick: ligand, green cube: Ni or Co ion). **b** Comparison of gas reactivity for MHHTP·$y$C/CuHHTP-5C (M=Ni, Co) with various gases: EtOH, TMA, NH₃, and NO₂. All results represent the average values from three different sensors, and the error bars indicated the standard deviation. All results represent the average values from different sensors (n = 2–4). Source data are provided as a Source data file.

sectional and top surfaces (Supplementary Figs. 9 and 10). The elemental mapping of SEM images revealed that the cross-section of the base films contains Cu, while the top overlayer surfaces contain Ni and Co, respectively. Furthermore, the XPS spectra confirmed the presence of Ni and Co in the NiHHTP-7C/CuHHTP-5C and CoHHTP-7C/CuHHTP-5C, respectively (Supplementary Figs. 11 and 12). The gas-sensing characteristics of MHHTP·$y$C/CuHHTP-5C (M = Ni, Co, $y$ = 1, 3, 5, 7) sensors were investigated in Fig. 3b. The gas experiment conditions were the same as previously described, where the sensor was stabilized under dark conditions before gas injection for 30 min (Supplementary Figs. 13 and 14). The responsiveness to the analytes varied depending on the metal sites and coating numbers.

In most sensing cases, the overlayer coatings either decreased or maintained the responses of the base CuHHTP-5C sensors; for example, the slight reduction in NH₃ responses was attributed to the

filtering effect caused by the increased overall film thickness after applying the overlayer coatings. However, across all coating cycles, Co₃HHTP₂ and Ni₃HHTP₂ showed increased sensitivities to TMA and NO₂, respectively. Although the sensitivity to NO₂ is enhanced by coating the Ni₃HHTP₂ layer, the key issue—lack of reversibility—still remained unresolved, leading to their exclusion from further consideration for sensor arrays (Supplementary Fig. 15). In contrast, the Co₃HHTP₂ coating showed significantly higher sensitivity to TMA than CuHHTP-5C; the TMA response of the CoHHTP-7C/CuHHTP-5C sensor was 12.5%, which is 73.9% higher than that of the CuHHTP-5C sensor.

The affinity of TMA for cobalt metal clusters has been previously demonstrated in examples such as cobalt-imidazole framework ([Co(im)₂]ₙ, im = imidazole) sensors[47]. This enhanced reactivity to TMA gas with the Co₃HHTP₂ overlayers was further confirmed through DFT calculations (Fig. 4). TMA is more likely to bind with metal cluster (MO₄, M = Cu, Ni, Co) than to triphenylene ligands, with two preferred binding modes: one involving amine-metal interactions and the other involving methyl-oxygen interactions. Among all combinations, the Co₃HHTP₂ exhibited the highest binding energy with the amine group of TMA at −1.09 eV, while the binding energies for Cu₃HHTP₂ and Ni₃HHTP₂ were lower, at −0.72 eV and −0.75 eV, respectively. For methyl-oxygen interactions, the binding energies were relatively low across all MOFs, ranging from −0.33 eV to −0.37 eV, presumably due to the reduced influence of metal sites. In addition, a comparison of the $Cu^{2+}/Cu^+$ ratio between CuHHTP-5C (Supplementary Fig. 3) and CoHHTP-7C/CuHHTP-5C (Supplementary Fig. 12) showed that $Cu^{2+}$ was reduced to $Cu^+$ after the Co₃HHTP₂ overlayer coating. This suggests that the electrical signal variation in the Co₃HHTP₂ layers induced by TMA adsorption can be transferred to the bottom Cu₃HHTP₂ layers, and changes the overall resistance of the film and enhancing its sensitivity. With regard to EtOH, NH₃, and NO₂ gases, DFT calculations were not conducted because their responses did not show noticeable improvement upon overlayer coating. Furthermore, unlike TMA, which exhibits consistent responses regardless of film thickness (Fig. 2d), the responses to EtOH, NH₃, and NO₂ are strongly influenced by the thickness of the films. Therefore, it is challenging to decouple the effects of film thickness from the catalytic effects of the overlayers in these cases.

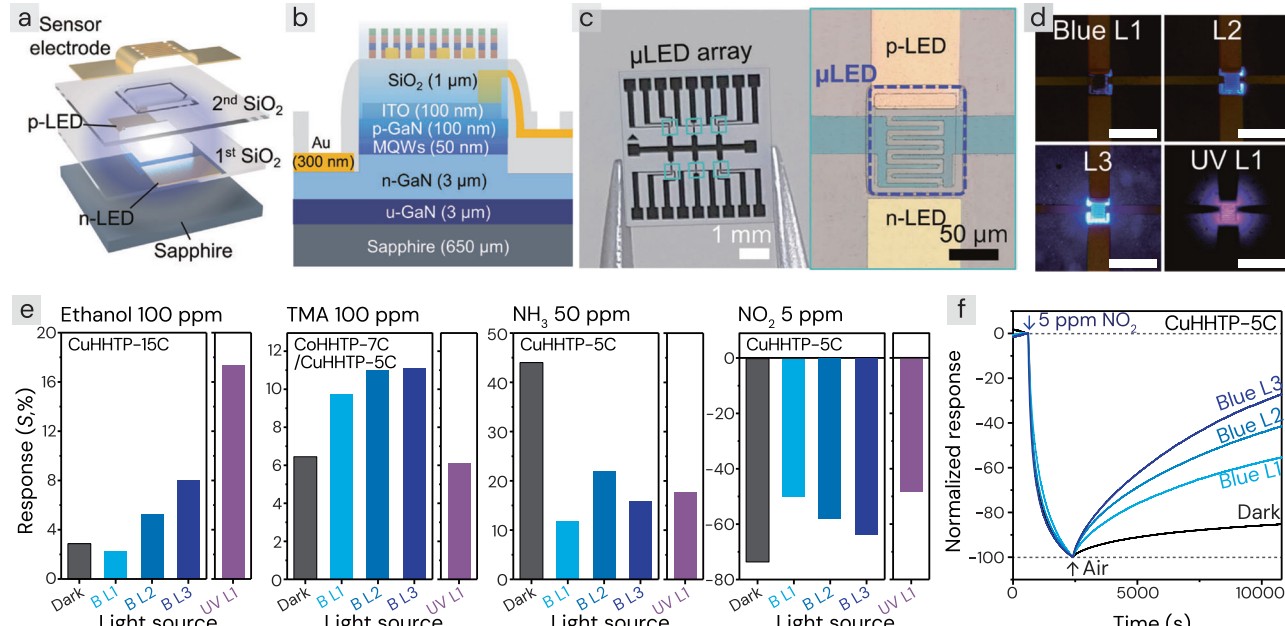

**Fig. 5 | cMOF gas sensor integrated on µLED platforms (µLP). a** 3D illustration and **b** cross-sectional schematic of the layered structure of the cMOF-integrated µLED sensor. **c** Optical microscopy image of the µLED sensor. **d** µLEDs emitting under varying light intensities and wavelengths. (Scale bar: 200 µm) **e** Response of the cMOF-µLED sensor under varying light intensities. **f** Normalized response of CuHHTP-5C integrated blue µLED to 5 ppm NO₂ under varying light intensities. Measurements were performed under five conditions: in the dark with the µLED turned off (Dark), under blue light at three different intensities (B L1: 63.2 µW, B L2: 260.4 µW, B L3: 864.6 µW), and under UV light (U L1: 66.1 µW). Source data are provided as a Source data file.

Based on the experiments on the sensing characteristics according to the configuration and composition of the cMOF sensing films, we were able to determine the optimal conditions for high responsiveness to each gas. For EtOH, a thicker sensing film was advantageous as it increased the number of reactive sites (e.g., CuHHTP-15C). In the case of TMA, the sensitivity improved when the Co₃HHTP₂ overlayer was coated (e.g., CoHHTP-7C/CuHHTP-5C). For NH₃ and NO₂ gases, a thinner sensing film was beneficial for enhancing sensitivity by avoiding filtering effects (e.g., CuHHTP-5C). It should be noted that sensors with fewer than 5 coating cycles were excluded, despite showing higher responses to NH₃ and NO₂, as the incomplete coatings of the sensing films resulted in sample deviations greater than 10% (Supplementary Table 1). We anticipate that gas sensor arrays can be fabricated using these optimized combinations of cMOF thin films. Furthermore, by integrating these combinations with a µLED substrate, accurate gas detection became feasible with enhanced sensitivity and selectivity of MOF chemiresistors, as well as high reversibility for continuous use in practical e-nose device. Utilizing various cMOF combinations can enhance sensitivity to a wide range of target gases, improving selectivity and increasing detection accuracy in the development of deep learning-based e-noses. Achieving reversibility ensures that the sensor can be used continuously and repeatedly, making it more durable and practical for long-term applications.

**Integration of cMOFs and µLP**

The optimized cMOF films (CuHHTP-5C, CuHHTP-15C, and CoHHTP-7C/CuHHTP-5C) were integrated with µLP by LBL coating methods, forming monolithic photoactivated cMOF-µLED array sensors. The µLEDs were produced by controlling the MOCVD process conditions of the gallium nitride (GaN) layers, varying the number of layers and their composition to achieve different light wavelengths. In this study, two µLEDs with wavelengths of 395 nm (UV) and 455 nm (blue) were used, providing photon energy sufficient to photoactivate the energy gap of the cMOFs. The µLPs were fabricated using micro-electro-mechanical system (MEMS) processes, insulated with a double-layered SiO₂, and

patterned with interdigitated electrode (IDE). When the cMOF is integrated on the IDE, the distance between the µLED light source and the material is 1 µm, enabling efficient transfer of light energy from the µLED (Fig. 5a, b). The detailed fabrication process of the µLP, along with the integration of the cMOF layer on top, is provided in Supplementary Fig. 16. Figure 5c presents an optical microscopy image of the actual fabricated µLED sensor. Multiple µLEDs are arranged in an array on a 5 × 5 mm² sensor chip, with each µLED measuring 100 × 100 µm². When a forward bias is applied, with a positive voltage to the p-electrode and a negative voltage to the n-electrode, the µLEDs emit light, transferring energy to the cMOF films above. The electrical resistance of the cMOFs was then measured via the IDEs.

Photoactivated gas sensors require optimal light intensity to effectively activate the gas-sensing material, as both insufficient and excessive light energy can negatively impact sensitivity. To determine the optimal conditions, gas-sensing tests were conducted under various lighting scenarios: in the dark condition with the µLED in the OFF state, under UV light (L1: 66.1 µW), and under blue light at three different intensities (L1: 63.2 µW, L2: 260.4 µW, and L3: 864.6 µW). For the UV µLED, since it was observed that prolonged use at high intensity can cause damage to the conductivity of the MOF, the experiment was conducted only at the L1 level. Nevertheless, even after prolonged UV L1 exposure for over 60 h, the cMOF maintained its conductivity and EtOH sensing performance (Supplementary Fig. 17), and Raman spectroscopy confirmed that no significant ligand degradation occurred after the UV exposure (Supplementary Fig. 18). The detailed light-current-voltage (L-I-V) properties of the µLED are summarized in Supplementary Fig. 19, and the actual emission of the µLED is shown in Fig. 5d. The responses of optimized cMOF films to each gas varied based on the operating conditions (wavelength and light intensity) of the µLED (Fig. 5e and Supplementary Fig. 20). The experimental results showed that for non-adsorptive EtOH and weakly adsorptive TMA, the response increased with increasing light intensity. For the EtOH responses of CuHHTP-15C sensors, the UV light at L1 intensity showed the highest responses due to the generation of photoactivated charge

carriers, which facilitated the reactions with neutral gas species. However, the highest TMA responses from the CoHHTP-7C/CuHHTP-5C sensors were observed under blue light, rather than UV light. This is presumably because the high-energy UV light not only increases the reaction sites for the neutral methyl group but also accelerates the desorption of basic amine sites of TMA gases. In contrast, when sensing basic $NH_3$ and acidic $NO_2$ using CuHHTP-5C sensors, the forward reaction is strongly favored, as shown in Eqs. (1) and (2) (Supplementary Fig. 20). In particular, $NO_2^-$ adsorbates are irreversibly bound to the MOF surfaces. At this stage, light illumination can generate electron–hole pairs, thereby accelerating the reverse reaction and promoting the desorption of adsorbates.

$$NH_{3(gas)} \leftrightarrow NH_3^+{}_{(ads)} + e^- \qquad (1)$$

$$NO_{2(gas)} \leftrightarrow NO_2^-{}_{(ads)} + h^+ \qquad (2)$$

Although hindrance of the forward reaction can limit gas sensitivity, light activation is beneficial in the case of $NO_2$, significantly improving the recovery rate and thereby enhancing reversibility (Fig. 5f). Considering that most MOF chemiresistors suffer from irreversibility to $NO_2$[48], their practicality was improved through a high recovery rate. Comparison under thermal and photoactivation conditions confirmed that photoactivation more effectively promotes $NO_2$ recovery of the cMOF sensor, demonstrating the superiority of the μLED-integrated cMOF sensor (Supplementary Fig. 21). In contrast, while $NH_3$ also shows slightly improved recovery with light activation, the corresponding decrease in sensitivity outweighs the benefit of faster recovery, leading to the conclusion that operating in dark conditions is optimal for $NH_3$ detection. Additionally, for TMA sensing using the CoHHTP-7C/CuHHTP-5C sensors, the L3 intensity was excluded from consideration. Despite consuming 3.3 times more power than L2 (864.6 μW vs. 260.4 μW), the TMA responses at L3 and L2 were similar, indicating no significant improvement in sensitivity.

The sensors were selected based on optimal conditions in terms of response magnitude and reversibility as follows: for EtOH gases, operating CuHHTP-15C with the UV at L1 intensity (sensor 1); for TMA gases, CoHHTP-7C/CuHHTP-5C with the blue at L2 intensity (sensor 2); for $NH_3$ gases, CuHHTP-5C with the μLED turned OFF (sensor 3); and for $NO_2$ gases, CuHHTP-5C with the blue at L2 intensity (sensor 4). Sensor 1 showed a response ($S$) of 17.3% to 100 ppm EtOH, sensor 2 exhibited $S = 11.0\%$ to 100 ppm TMA, sensor 3 demonstrated $S = 44.0\%$ to 50 ppm $NH_3$, and sensor 4 displayed $S = 57.9\%$ to 5 ppm $NO_2$. The total power consumption of the four-sensor array was 587 μW, keeping it below 1 mW. With their optimal sensitivity, all sensors were reversible, enabling repeated and continuous gas detection with minimum power consumptions. Cycle tests under these optimized conditions (Supplementary Fig. 22) confirmed good repeatability, with μLED photoactivation aiding recovery of the cMOF sensors. As such, the μLED-cMOF sensor platform offers a highly effective approach for designing sensors involving diverse gas species. By combining the ability of μLEDs to freely adjust emission wavelength and intensity with the tuning of LBL-based MOF sensing films, it becomes possible to achieve optimal operating conditions in new applications.

The ultraviolet-visible (UV-vis) spectra of $Cu_3HHTP_2$ films showed an absorbance peak at 360 nm, attributed to the π−π* transition of the HHTP ligands, alongside a broad peak near 645 nm resulting from ligand-to-metal charge transfer, which imparts room-temperature conductivity to the metal-organic frameworks (MOFs) (Supplementary Fig. 23). The energy gaps of the π−π* transition states were calculated to be 2.90 eV for CuHHTP-5C and 2.86 eV for CuHHTP-15C, suggesting that they can be excited by μLEDs illuminations (3.13 eV for UV LED and 2.72 eV for blue LED). The CoHHTP-7C films exhibited a similar energy gap of 2.94 eV due to the involvement of the same HHTP ligands.

To confirm that the sensing mechanism of the μLED-integrated cMOF sensor is driven purely by photoactivation rather than photothermal-induced temperature increase, we employed an infrared micro-thermography system (Supplementary Fig. 24). Temperature measurements were conducted while operating the blue μLED under a forward bias ranging from 1 V (LED OFF) to 4 V, for both the blue μLED-only sample and the blue μLED with the $Cu_3HHTP_2$. Under continuous blue μLED (L2) illumination for 60 min, both samples exhibited minimal temperature changes of less than 0.5 °C (Supplementary Fig. 25). This finding confirms that the gas sensing of the cMOF is facilitated by the activation of its energy gap through photoactivation by the μLED, rather than by any temperature increase.

## Deep learning-based cMOF e-nose system

An e-nose system was developed using the cMOF array to selectively distinguish between different gases. The e-nose employed an array of four sensors (sensors 1–4) optimized for detecting target gases such as EtOH, TMA, $NH_3$, and $NO_2$, along with CNN-based pattern recognition for classifying gas types and predicting concentrations through regression (Fig. 6a). Training data were collected simultaneously from the sensor array for various concentrations: EtOH (50, 100, 200 ppm), TMA (50, 100, 200 ppm), $NH_3$ (10, 20, 50 ppm), and $NO_2$ (1, 2, 5 ppm) at a sampling rate of 1 s. To enhance the accuracy of the deep learning model, data augmentation was performed. Since there was less than a 10% variation in gas response across same gas experiments (as detailed in Supplementary Table 1), a ±10% deviation was applied to the sensor signals in the response and recovery phases of the original gas experiment data (Supplementary Fig. 26). In summary, after data augmentation, three sets of gas experiment data ($R_{original}$, $R_{+10\%}$, $R_{-10\%}$) were used for deep learning. The sensor array signals were normalized to $R_g/R_a$, then concatenated in the form of a 4 (number of sensors) × time (s) matrix, and a sliding time window of $4 \times 60$ s was applied. During the training of the CNN model, the gas response data were combined with their corresponding true labels (gas type and concentration). For gas-type classification, the true labels were assigned as follows: air, EtOH, TMA, $NH_3$, and $NO_2$ were labeled as 0, 1, 2, 3, and 4, respectively. For gas concentration regression, to prevent unwanted bias caused by the different concentration ranges of each target gas, normalization was performed by dividing the gas concentration by the maximum concentration of each target gas, resulting in values between 0 and 1. The detailed structure, hyperparameters, and training process of the deep learning model are depicted in Supplementary Fig. 27.

The average accuracy for classifying normal air and the four target gases was 99.8%, while the mean absolute error (MAE) for concentration prediction was 7.94%. Figure 6b shows the real-time classification of gas types (represented by black dots) and concentration regression (colored dots corresponding to each gas), with the red dotted line indicating the true concentration of the injected gas. Figure 6c presents the confusion matrix for classification results, and Fig. 6d shows the regression results for the normalized gas concentrations. The prediction errors for gas types and concentrations are shown in Supplementary Table 2, with the largest error observed for TMA at 50 ppm (22.5%) due to the sensor's lowest response. In contrast, the errors for the other gases were approximately 10%. Additionally, although the response and recovery times of the cMOF sensors are on the order of tens of minutes (Supplementary Fig. 20), the CNN-based algorithm, even when considering the 60-s sliding time window, can predict gas types and concentrations within 2 min, making it highly practical for real-time applications. To the best of our knowledge, this is the first example of a cMOF-based e-nose system capable of recovery, allowing for repeated use.

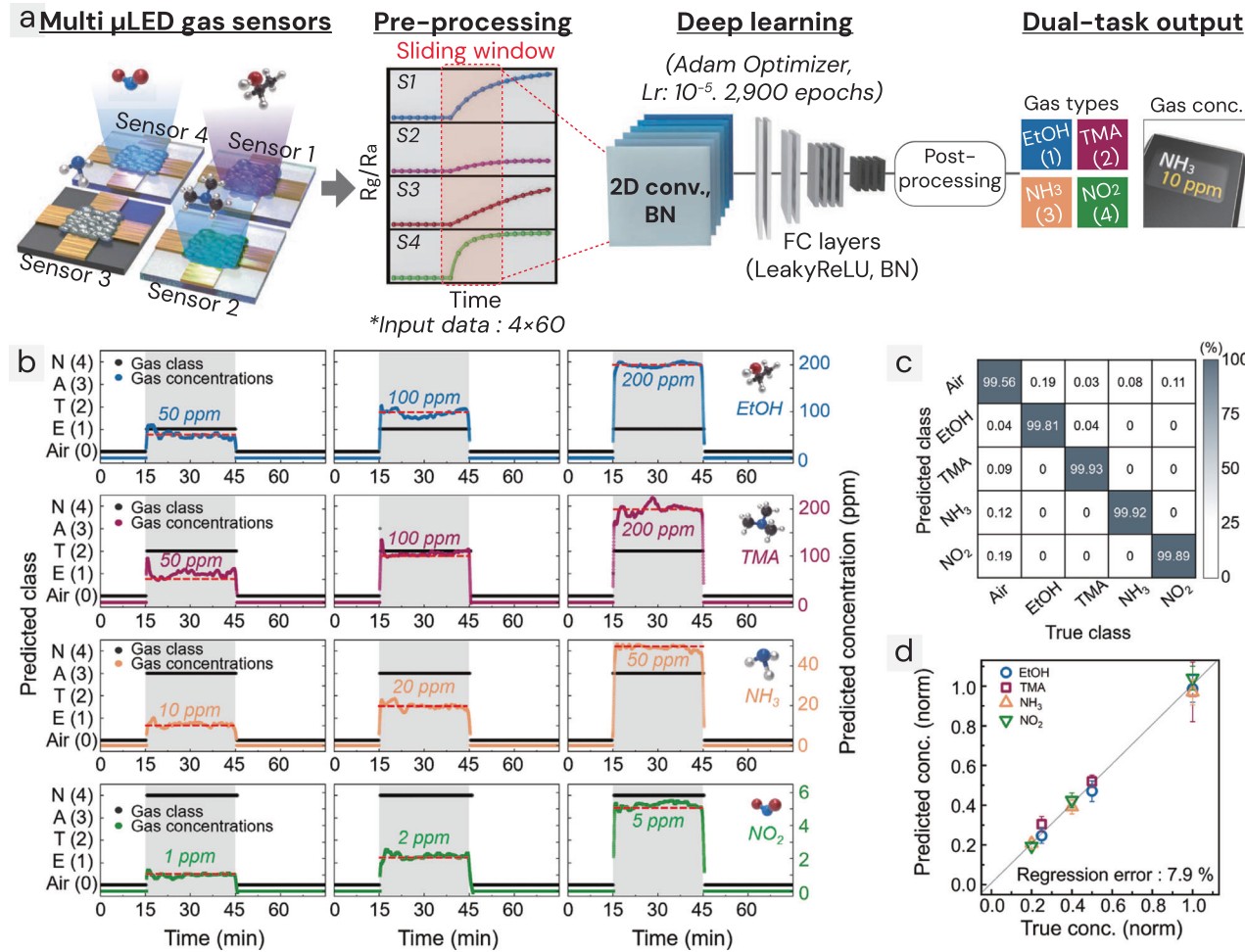

**Fig. 6 | cMOF-based chemiresistive sensor arrays and deep learning-based e-nose system. a** Structure of the e-nose model using four different cMOF sensors and CNN-based pattern recognition to distinguish gas types and predict concentrations. Lr denotes the learning rate, BN refers to batch normalization, which are ML hyperparameters, and LeakyReLU (leaky rectified linear unit) is an activation function. **b** Real-time prediction results for 4 different gases: EtOH (E), TMA (T), $NH_3$ (A), and $NO_2$ (N). **c** Confusion matrix showing classification results.

**d** Regression results for concentration prediction (normalized from 0 to 1). Error bars represent the mean ± standard deviation of 1800 ML predictions generated from repeated time-resolved measurements collected by a single biological sensor during gas exposure. While the input data include biological replicates, the plotted prediction results themselves are not replicated. Source data are provided in the Source data file.

## Discussion

This study successfully prepared highly sensitive gas sensor arrays using LBL methods to coat various cMOF films and integrate them onto MEMS-fabricated µLPs. First, by understanding the relationships between the thickness of $Cu_3HHTP_2$ films and the adsorption properties of gases, we optimized the sensitivity for highly adsorptive acidic $NO_2$ and basic $NH_3$ by using thinner films to reduce gas filtering effects, while increasing the thickness for non-adsorptive neutral EtOH gases to enhance the number of gas reactions. For weakly adsorptive TMA, which showed no thickness dependency, we employed additional $Co_3HHTP_2$ overlayers, as the high binding energy between the amine group and Co metals contributed to enhanced sensitivity, as confirmed by DFT calculations. Additionally, the illumination of cMOFs with µLEDs generated photoactivated charge carriers that increased sensitivity to EtOH and TMA, and effectively addressed the issue of irreversible $NO_2$ sensing, facilitating repetitive gas sensing. By optimizing the types of µLEDs and their light intensities for corresponding cMOF films, accurate detection of four target gases was achieved with a total power consumption of 587 µW, showcasing its potential for widespread, energy-efficient applications. Furthermore, the system was combined with a CNN-based deep learning algorithm, enabling a dual-task e-nose capable of distinguishing between different gases and

predicting their concentrations with high accuracy in real-time. The system achieved a classification accuracy of 99.8% and a regression error of 7.94%. This highly precise and selective gas detection technology holds great potential for future applications, including IoT-based systems, environmental monitoring, and industrial safety. Although there are currently few types of cMOFs that exhibit high conductivity when fabricated as thin films, the limitless versatility of MOFs suggests that the combination of MOF-based chemiresistor arrays will expand dramatically in the near future. This advancement will pave the way for the practical realization of e-noses devices.

## Methods

### Preparation of $Cu_3HHTP_2$ sensing films

Au-interdigitated silicon substrates and µLED substrates were treated with UV/ozone for 10 min to create hydrophilic surfaces. For the LBL coating process, a 1.0 mM ethanol solution containing copper acetate hydrate and a 0.2 mM ethanol solution containing $H_6HHTP$ were prepared as the metal and ligand solutions, respectively. During each cycle, the substrates were immersed in the metal solution for 1 min, followed by immersion in the ligand solution for 2 min, with a wash in pure ethanol between each step to remove any residual reagents.

### Preparation of M₃HHTP₂ (M = Ni, Co) overlayers on Cu₃HHTP₂ sensing films

A 0.1 mM ethanol solution containing metal salts and a 0.02 mM ethanol solution containing $H_6HHTP$ were prepared as metal and ligand solutions, respectively. The as-prepared sensors (e.g., CuHHTP-5C) were immersed in the metal solution for 1 min, followed by immersion in the ligand solution for 2 min, and washed with pure ethanol between each step to remove any residual reagents.

### Fabrication of sensor devices without µLED illumination

$SiO_2$ insulation layer was deposited on a Si wafer using plasma-enhanced chemical vapor deposition (PECVD), followed by photo-lithography and e-beam evaporation to pattern a 10 nm Cr and 200 nm Au IDE with the same dimensions as the µLED sensor. The wafer was then diced into $5 \times 5$ mm² sensor chips using a blade dicing process (Supplementary Fig. 1).

### Gas-sensing experiments of non-emissive sensors

The sensing characteristics of non-emissive sensors, without integrated µLEDs, were investigated to optimize the MOF film conditions. The gas-sensing chamber was equipped with two mass-flow controllers (MFCs), each with a constant flow rate of 1000 sccm. One MFC was connected to dry air for sensor stabilization, and the other was connected to the analyte gases to establish the desired gas atmosphere. A 4-way valve was used to switch between the two gases for instantaneous injection. The two-probe DC resistance of the sensors was measured using a DAQ970A multimeter (Keysight), with data acquisition handled by a computer.

### UV exposure and Raman analysis of cMOF films (CuHHTP-15C)

Continuous UV irradiation was performed using a 395 nm wavelength lamp (110 lm, 3 W), positioned 5 cm above the sensor. The cMOF film (CuHHTP-15C) was exposed to the UV light for over 60 h. Raman spectroscopy was conducted before and after the UV exposure to monitor any potential changes in the molecular structure.

### Fabrication of micro-LED sensor platform

A 3 µm thick n-GaN layer, a 3 µm thick undoped GaN layer, a 50 nm InGaN/GaN multi-quantum-well (MQW) active layer, and a 100 nm p-GaN layer were grown epitaxially on a 650 µm thick sapphire substrate using metal-organic chemical vapor deposition (MOCVD) (Outsourced to Soft-epi, Korea). The GaN layers were vertically etched using inductively coupled plasma-reactive ion etching (ICP-RIE) to create a $100 \times 100$ µm² mesa structure. An indium tin oxide (ITO) current-spreading layer was deposited via e-beam evaporation, followed by rapid thermal annealing (RTA) to enhance transparency and electrical conductivity. Gold $p$ and $n$ contact electrodes were formed through photolithography and e-beam evaporation. A first $SiO_2$ insulation layer was deposited by PECVD and etched using reactive ion etching (RIE) to expose the $p$ and $n$ contacts. A second $SiO_2$ layer was also deposited via PECVD and etched using RIE to re-open the contact pads. Gold IDEs were then patterned on the µLED platform through photolithography and e-beam evaporation. Finally, the fabricated µLED gas sensors were diced into $5 \times 5$ mm² chips using blade dicing (Sapphire dicing blades, Disco, Japan).

### Gas-sensing experiments of µLEDs integrated sensor arrays

The µLED gas sensor was assembled on a printed circuit board (PCB), followed by gold wire bonding to establish connections. The sensor-embedded PCB was then housed in a custom-built polycarbonate (PC) enclosure and linked to a dual-channel sourcemeter (2636B, Keithley, USA), which enabled individual voltage control for each µLED and simultaneous resistance measurement of the sensor (bias: 1 V). The MFC (AFC500, ATOVAC, Korea) was used to regulate the precise flow of dry target gases (0% relative humidity) such as EtOH, TMA, $NH_3$, and $NO_2$, which were then introduced into the PC enclosure.

### Density functional theory (DFT) calculations

DFT calculations were carried out using the Vienna Ab Initio Simulation Package (VASP) version 5.4.1[49]. The monolayer structure of M₃HHTP₂ (M = Co, Ni, Cu) was modeled with a 20 Å vacuum slab, which was adapted from prior research[50]. The Perdew–Burke–Ernzerhof (PBE) functional within the generalized gradient approximation (GGA) was used in conjunction with the projector augmented wave (PAW) method. A plane-wave energy cutoff (ENCUT) of 520 eV was applied, which is 1.3× higher than the default ENMAX of the pseudopotentials, and the precision level was set to Accurate (PREC). Spin-polarized calculations (ISPIN = 2) were employed to capture magnetic effects, and a Gaussian smearing method (ISMEAR = 0) with a width of 0.05 eV (SIGMA) was used for electronic occupations. To account for van der Waals interactions between the TMA molecule and the MOF layer, Grimme's D3 dispersion correction with zero damping (IVDW = 11) was included[51,52]. Convergence criteria for electronic and ionic optimizations were set at $10^{-5}$ eV (EDIFF) and 0.003 eV/Å (EDIFFG), respectively. The optimizations were performed using the conjugate gradient algorithm (IBRION = 2), with no symmetry constraints imposed (ISYM = 0). The k-points sampling was performed using a gamma-centered $1 \times 1 \times 1$ grid within the Brillouin zone. The binding energy of the TMA molecule with the MOF was calculated using the formula $E_{binding} = E_{MOF+TMA} - E_{MOF} - E_{TMA}$.

### Machine learning environment and computational resources

The CNN algorithm was implemented using the open-source machine learning library PyTorch (Meta, USA). Model training was accelerated in a high-performance computing environment equipped with an RTX Titan GPU (NVIDIA, USA).

### Reporting summary

Further information on research design is available in the Nature Portfolio Reporting Summary linked to this article.

## Data availability

All data supporting the findings of this study are available within the article and its supplementary files. Any additional requests for information can be directed to, and will be fulfilled by, the corresponding authors. Source data are provided with this paper.

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

## Acknowledgements

I.P. was supported by the National Research Foundation of Korea (NRF) grant funded by the Korean government (MSIT) (No. 2021R1A2C3008742). Y.C.K. was supported by the National Research Foundation of Korea (NRF) grant funded by the Korean government (MSIT) (No. 2021M3H4A3A02086430). K.S.C. was supported by the Korea Basic Science Institute grant (No. D537200). The authors are deeply grateful to the late Professor Jong-Heun Lee for his wisdom and passion for the field, which have left a lasting influence. This work reflects the inspiration the authors drew from his invaluable guidance.

## Author contributions

K.L., Y.M.J., and M.S.S. contributed equally to this work. K.L., Y.M.J., and M.S.S. planned and conducted the experiments and wrote the paper. M.J. and J.K. contributed to the DFT calculations. C.K. assisted with material analysis and data organization. O.G., S.J.P., and K.B.K. contributed to the review and editing. K.S.C. and C.B.J. contributed to the experiments and analysis for characterizing the surface temperature of the cMOF and μLP. Y.M.J., Y.C.K., and I.P. supervised the project. All authors have given approval to the final version of the manuscript.

## Competing interests

The authors declare no competing interests.
