## [Transparent Peer review file · Nature Communications]

Photoactivated Conductive MOF Thin Film Arrays on Micro-LEDs for Chemiresistive Gas Sensing

Corresponding Author: Professor Inkyu Park

Version 0:

Reviewer comments:

Reviewer #1

(Remarks to the Author)

This study prepared uLED-assisted cMOF or cMOF-on-cMOF thin film gas sensor arrays using LBL methods and revealed the relationships between the thickness/composition of cMOF films and the photoactivated gas sensing properties. Overall, the results are interesting and meaningful for guiding the practical realization of cMOF based e-noses devices. The paper is organized well and the language is fluent. However, the mechanisms of the cMOF thin film growth and devices still need further improvement before it can be accepted for publication in Nat Commun. More comments are listed below:

1. Abstract, Line 18-19: what is the advantage of integrating cMOF onto u-LED? Why did you do that? What is the relationship between room-temperature operation limits and integrating cMOF onto u-LED? Such questions will be raised by readers on reading current version of the abstract. In the introduction and the results, we can see its advantage on the good recovery toward NO₂, I suggest the authors to revise this part for a reader-friendly abstract.
2. Line 47-49: any reference for this statement?
3. Line 116- 117, did you conduct AFM analysis to measure the thickness of the films? AFM results are the solid proof of the thickness of thin films. In addition, the roughness of Cu-HHTP thin film fabricated by the method used in this work is too high, and the surface shown in the SEM we can even observe significant particles. In your further work, please also consider to develop a better strategy to improve the quality of the thin film, which will also contribute to high performances.
4. Line 139- 140, according to the Supplementary Fig.4, it is difficult to identify if the Cu₃HHTP₂ films completely cover the substrate or not before 5 cycle coatings.
5. Line 143- Line 145, what is the difference between the in-plane and out-of-plane growth of Cu₃HHTP₂. How could the authors distinguish it according to the SEM images?
6. Line 145- 147, is there any reference to support this statement? It is correct that connections are very important for the resistance decrease. But if the authors check the conductivity of the thin film, which excludes the effect of thickness, they will figure out that defects decrease during the LbL growth is also a very important factor. Please be careful about such statements without solid proofs.
7. Line 170-171, why do cMOFs with a light source enhance the repeatability? Could you explain more about this statement (for instance, the improved recovery)?
8. Line 229- 230, did you conduct any characterizations for the film to support this statement? Ni-HHTP or Co-HHTP is very difficult to form a crystalline thin film by such methods, it is quite difficult to tell how they connect if they are just amorphous coordination polymers without periodic structures.
9. Line 259, please add the full name of IDE?
10. Line 292-284, as mentioned in comment 1, for a reader-friendly version, the authors are recommended to clarify the following question: will the light activation affect the NO₂ or NH₃ gas adsorption and desorption on the film? You'd better take this influence into consideration when you analyze why light activation enhanced reversibility. Why light activation enhanced reversibility for NO₂ and NH₃ gas needs deep analysis and discussion.

Reviewer #2

(Remarks to the Author)

In this manuscript Park et al. sophisticatedly designed and prepared conductive MOF thin film arrays on micro-LEDs for gas sensing. Comprehensive characterizations and investigations were conducted on this system to demonstrate the work

mechanism. A so-called e-nose technology was developed for gas detection. Basically I think it holds the novelty and significance for Nat. Commun. The manuscript is well written. It could be acceptable if the following issues could be addressed:

1. More discussion should be given to clarify why the four specific gases (EtOH, TMA, NH₃ and NO₂) were chosen. What kind of application scenarios is this gas sensing technology used for? What is the challenge?
2. For the c-MOF, why this MHHTP was selected? During the fabrication of MOF overlayers, it is quite possible that the lattices will merge with each other to form a mixed metal layer, as shown in Fig. 8d-e. Is there any effect on the sensing performances? Please clarify this.
3. The deep-learning is difficult to understand. More introduction and discussion should be provided. After the deep-learning, what have been achieved as great improvements.
4. In practical applications, gas sensing is usually used in complicated gas mixture. The potential of such applications should be conducted.
5. Except for the NO₂ gas, other gases, as they stated, are reversible. How is about the recyclability? I suggested collecting the experimental data, showing the sensing performances after tens of times.
6. Some typos: L162, "It presumably resulting in intermediate response trends between 163 non-adsorptive gas and highly adsorptive gases (e.g., NH₃).". Ref., 12, 22, 29, 30, 39, etc.

Reviewer #3

(Remarks to the Author)

This study presents a novel integration of photoactivated conductive metal-organic framework (cMOF) thin films with micro-LEDs (μ LEDs) for chemiresistive gas sensing, achieving ultra-low power consumption (587 μ W) and high classification accuracy (99.8%) using deep learning. The systematic optimization of cMOF film thickness, overlayer composition, and μ LED illumination parameters demonstrates a compelling approach to address challenges in room-temperature gas sensing, such as reversibility and selectivity. The combination of experimental data, DFT calculations, and CNN-based pattern recognition provides a robust framework for advancing e-nose technology. However, several methodological details, data interpretations, and technical validations require clarification to strengthen the claims and ensure reproducibility.

1. In Fig. 2d, the response trends for NH₃ and NO₂ decrease with increasing CuHHTP-xC cycles, while EtOH responses saturate. What is the quantitative relationship between film thickness (Supplementary Fig. 4) and gas adsorption kinetics? Provide diffusion coefficients or adsorption energy calculations to support the "gas filtering effect" hypothesis.
2. On Page 7, Lines 148–153, The irreversible recovery of NO₂ (Supplementary Fig. 6) is attributed to strong adsorption. However, the mechanism of light-activated reversibility (Fig. 5f) is not quantitatively compared to thermal desorption. Include temperature-dependent recovery experiments under dark/light conditions to isolate photochemical vs. thermal effects.
3. In Fig. 3b, CoHHTP-7C/CuHHTP-5C shows a 73.9% increase in TMA response. Clarify whether this enhancement is solely due to Co's binding energy (Fig. 4) or influenced by charge transfer between Co/Cu layers. Provide XPS or UPS data to confirm interfacial electronic interactions.
4. In Fig. 5e, the EtOH response under UV light (L1) is higher than under blue light. However, UV photons (3.13 eV) exceed the π - π^* transition energy (~2.9 eV). Does this cause ligand degradation? Include FTIR or Raman spectra of UV-exposed films to confirm structural stability.
5. On Page 15, Lines 315–318, The UV-vis spectra show a broad peak at 645 nm (ligand-to-metal charge transfer). How does this align with the μ LED wavelengths (395 nm, 455 nm)? Provide band diagrams to illustrate photoexcitation pathways.
6. In Supplementary Fig. 19, the temperature rise under μ LED illumination is <0.5°C. Were measurements taken under continuous operation (e.g., 1 hour)? Include long-term stability data to rule out cumulative heating effects.
7. In Fig. 2h, the NH₃ response of CuHHTP-1C (457.1%) is claimed superior to prior cMOF sensors. Compare response times (t_{90}) with literature to assess practicality for real-time applications.
8. On page 14, Lines 294, in Fig. 5f, NO₂ recovery rate improves under blue light (L2). Quantify the recovery rate for repeated cycles to validate long-term reversibility.
9. On Page 18, Lines 364–365, the CNN predicts gas types within "tens of seconds." Specify the exact latency (e.g., 20 s vs. 60 s) and its dependence on sensor response kinetics.
10. On Page 11, Lines 229–232, the Co/Cu alloy phase is hypothesized but not confirmed. Perform TEM/SAED or XRD to verify epitaxial growth and lattice matching.
11. In Fig. 6d, The MAE for concentration regression is 7.94%. Does this error vary with gas type (e.g., higher for TMA)? Include error distributions for individual gases.
12. In Supplementary Fig. 17, sensitivity decreases for NH₃ under light. Does this correlate with hole/electron recombination? Provide photoluminescence quenching or other data to elucidate carrier dynamics.

13. In Fig. 4, DFT calculations focus on TMA binding. Extend simulations to NH₃ and NO₂ to explain selectivity trends in CuHHTP-xC sensors. Meanwhile, please further supplement the DFT calculation details, such as energy cutoff (ENCUT) and other parameter settings.

14. On Page 14, Lines 278–282, EtOH responses increase with light intensity. Is this due to photoactivated surface reactions or bulk conductivity changes? Perform impedance spectroscopy or other methods under illumination for further analysis.

15. On Page 17, Lines 342–343, data augmentation applies $\pm 10\%$ noise. Does this artificially inflate accuracy? Compare model performance with/without augmented data.

16. On Page 9, Lines 172–174, CuHHTP-1C and CuHHTP-3C show high NH₃ responses but are excluded due to deviations. Could a hybrid array (thick + thin films) improve overall performance?

Version 1:

Reviewer comments:

Reviewer #1

(Remarks to the Author)

The authors have addressed most of my questions. But there is still something unclear.

R1-Q3 and R1-Q4: Could you please explain why the roughness of the cMOF film does not significantly affect the gas sensing performance. It is risky to have such statements.

Reviewer #2

(Remarks to the Author)

All the issues I concerned have been well addressed. Now it could be accepted for publication.

Reviewer #3

(Remarks to the Author)

Accept as is.

Version 2:

Reviewer comments:

Reviewer #1

(Remarks to the Author)

While I disagree with the author's interpretation of the relationship between roughness and sensing properties—as the current evidence does not sufficiently substantiate their conclusion—this does not diminish the article's overall novelty. Therefore, I recommend its publication.

Dear Editor of *Nature Communications*:

We greatly appreciate the reviewers' comments. We agree with most of them and have included our point-by-point response to the reviewers' comments, which are listed as follows.

Reviewer #1 (Remarks to the Author):

This study prepared μ LED-assisted cMOF or cMOF-on-cMOF thin film gas sensor arrays using LBL methods and revealed the relationships between the thickness/composition of cMOF films and the photoactivated gas sensing properties. Overall, the results are interesting and meaningful for guiding the practical realization of cMOF based e-noses devices. The paper is organized well and the language is fluent. However, the mechanisms of the cMOF thin film growth and devices still need further improvement before it can be accepted for publication in *Nat Commun*. More comments are listed below:

We are very grateful to Reviewer #1 for the thoughtful comments and suggestions. In response to the reviewer's requests, we have made every effort to address each comment thoroughly. We have created correction boxes to clearly show the changes made in the revised manuscript, with both the original (before) and revised (after) texts provided for each modification.

*Please note that the page and line numbers indicated in each correction box correspond to the highlighted version of the revised manuscript, where all changes are marked for clarity.

[R1-Q1] Abstract, Line 18-19: what is the advantage of integrating cMOF onto μ LED? Why did you do that? What is the relationship between room-temperature operation limits and integrating cMOF onto μ LED? Such questions will be raised by readers on reading current version of the abstract. In the introduction and the results, we can see its advantage on the good recovery toward NO₂, I suggest the authors to revise this part for a reader-friendly abstract.

[R1-A1] Thank you very much for your comment. Currently, replacing high-temperature-operated metal oxide-based gas sensors with other room-temperature-operating materials has been a major focus to overcome the various disadvantages of high-temperature operation, such as high-power consumption, device degradation, and limitations in composition. Among many candidates, electrically conductive metal-organic frameworks (cMOFs) are one of the best materials that work at room temperature. Additionally, they offer a diverse library of structures simply by changing the composition of metals or linkers.

However, due to the low temperature (room temperature), the reaction speed is inevitably slow, and in some cases, complete recovery is not achieved. Irreversibility is a critical problem for practical applications; therefore, to resolve this issue without high power consumption or device degradation, the use of μ LEDs has been identified as the best approach. The μ LEDs, with a power of ~ 0.1 mW, enable room-temperature operation while simultaneously improving the reversibility and enhancing the sensitivity to specific gases. This approach can effectively address various challenges associated with conventional metal oxide gas sensors.

We agree that the current abstract lacks sufficient background information for the readership. Therefore, we have incorporated this information and refined the logical flow to enhance clarity.

<After Correction> Abstract (page 2, line 18)

(Original) Electrically conductive metal-organic frameworks (cMOFs) are emerging as promising chemiresistors due to their diverse compositions and high surface areas, enabling facile modification of chemical reactions. However, room-temperature operation limits sensitivity, selectivity and reversibility. In this study, cMOF thin films were integrated onto a micro-LED (μ LED) platform using a layer-by-layer method for photoactivated gas sensing.
→ Electrically conductive metal-organic frameworks (cMOFs) are emerging as promising chemiresistors due to their diverse compositions and high surface areas, enabling facile modification of chemical reactions. However, room-temperature operation limits sensitivity, selectivity and reversibility, **chemical properties, porosity, and room-temperature conductivity, enabling the design of energy-efficient devices. However, limited activation at room-temperature limits sensitivity and reversibility.** In this study, cMOF thin films were integrated onto a micro-LED (μ LED) platform using a layer-by-layer method ~~for photoactivated gas sensing~~, **enabling photoactivated gas sensing even at room-temperature.**

[R1-Q2] Line 47-49: any reference for this statement?

[R1-A2] Thank you for pointing this out. In the original version of the manuscript, lines 47–49 read as follows: "*While the development of light-activated MOF chemiresistor arrays is still in its early stages, the vast combinations of emerging MOFs and diverse light sources hold great promise for creating high-performance sensor arrays.*"

Many studies have already explored the fluorescent, or photocatalytic properties of insulating MOFs by modifying their optical band structures through the substitution of linkers or metal clusters (*Acc. Chem. Res.* 2020, 53, 2, 485–495, *Acc. Chem. Res.* 2019, 52, 2, 356–366). The relationship between the light source and the band structure of MOFs has been extensively studied. However, as these studies have primarily focused on insulating MOFs, their applicability to electronic devices has been limited.

In 2021, the current author was the first to propose the photoactivation of cMOFs, (*ACS Cent. Sci.* 2021, 7, 7, 1176–1182) and due to the lack of similar studies, it is evident that this field is still in its very early stages.

To provide further clarity, we have added relevant references and revised the sentence as

follows.

<After Correction> Introduction (page 3, line 52)

(Original) While the development of light-activated MOF chemiresistor arrays is still in its early stages, the vast combinations of emerging MOFs and diverse light sources hold great promise for creating high-performance sensor arrays.

→ While the development of light-activated MOF chemiresistor arrays is still in its early stages, studies on luminescent or photocatalytic properties have revealed various relationships between the band gap of MOFs and light^{21, 22}. These support the rationality of charge transfer induced by light. It is worth noting that applying light-activations on cMOFs has rarely been reported. Therefore, the vast combinations of emerging cMOFs and diverse light sources hold great promise for creating high-performance sensor arrays.

Ref 21, 22: *Acc. Chem. Res.* 53, 485–495 (2020), *Acc. Chem. Res.* 52, 356–366 (2020).

[R1-Q3] Line 116-117, did you conduct AFM analysis to measure the thickness of the films? AFM results are the solid proof of the thickness of thin films. In addition, the roughness of Cu-HHTP thin film fabricated by the method used in this work is too high, and the surface shown in the SEM we can even observe significant particles. In your further work, please also consider to develop a better strategy to improve the quality of the thin film, which will also contribute to high performances.

[R1-A3] We sincerely appreciate the reviewer's insightful comments. Unfortunately, due to the presence of agglomerated particles on the surface of the films, it is difficult to measure the thickness using AFM analysis (**Fig. R1**).

Fig. R1. AFM image of CuHHTP-5C on Si substrate.

However, we have provided SEM images that show a trend depending on the number of cycles (**Supplementary Fig. 4**). The thickness value was obtained by averaging ten repeated measurements, which supports the accuracy of this trend.

EDS analysis revealed that these unwanted particles are simply agglomerated Cu₃HHTP₂. Except C, Cu, O, and Si, other impurities are not found (**Figs. R2–R4**).

Fig. R2. **a** Top view SEM image of CuHHTP-1C on Si substrate. **b–e** EDS mapping of CuHHTP-1C on Si substrate; **(b)** C element, **(c)** Cu element, **(d)** O element, and **(e)** Si element. (Mass% of C: 4.57 ± 0.02 , O: 10.86 ± 0.03 , Si: 85.46 ± 0.10 , Cu: 0.01 ± 0.02)

Fig. R3. **a** Top view SEM image of CuHHTP-3C on Si substrate. **b–e** EDS mapping of CuHHTP-3C on Si substrate; **(b)** C element, **(c)** Cu element, **(d)** O element, and **(e)** Si element. (Mass% of C: 5.28 ± 0.02 , O: 10.78 ± 0.03 , Si: 83.77 ± 0.10 , Cu: 0.17 ± 0.02)

Fig. R4. **a** Top view SEM image of CuHHTP-5C on Si substrate. **b–e** EDS mapping of CuHHTP-5C on Si substrate; **(b)** C element, **(c)** Cu element, **(d)** O element, and **(e)** Si element. (Mass% of C: 6.31 ± 0.03 , O: 10.88 ± 0.03 , Si: 82.50 ± 0.09 , Cu: 0.30 ± 0.02)

The agglomerated particles result from the rapid reaction process, which involves immersing the substrate in the ligand solution for 2 minutes, followed by 1 minute in the metal solution, with tens of seconds for cleaning. Extending the reaction and cleaning time could improve surface roughness by completely removing residual salts at each step. However, if there is no batch-to-batch difference in device performance, increasing the process time may not be the optimal choice from an engineering perspective. We conducted multiple batches (2–4 times) for each sensor condition, and the variation remained minimal ($< 10\%$), indicating that this roughness does not significantly affect the gas sensing performance. Nevertheless, in future work, we will further refine the reaction and cleaning times to enhance the film structure. We have added **Figs. R2–R4** as **Supplementary Fig. 6** to provide additional visual evidence.

<After Correction> Supplementary Information

Supplementary Fig. 6 | a–c Top-view SEM image and EDS analysis of CuHHTP-1C, CuHHTP-3C, and CuHHTP-5C. (i) Top view SEM image of CuHHTP-5C on Si substrate. (ii–v) EDS mapping of CuHHTP-5C on Si substrate; (ii) C element, (iii) Cu element, (iv) O element, and (v) Si element. The particles on the film surfaces are confirmed as the agglomerated Cu_3HHTP_2 particles. These particles are grown through the rapid LBL process to optimize the engineering process, but this does not critically affect the gas sensing properties, as the batch-to-batch deviations remain below 10% (see Supplementary Table 1).

[R1-Q4] Line 139-140, according the **Supplementary Fig. 4**, it is difficult to identify if the Cu_3HHTP_2 films completely cover the substrate or not before 5 cycle coatings.

[R1-A4] Thank you for your comment. In the original version of the manuscript, lines 139–140 read as follows: “As evidence, the Cu_3HHTP_2 films did not completely cover the electrode before 5 cycle coatings (**Supplementary Fig. 4**).”

We were able to measure resistance and even conduct chemiresistive sensing tests with 1- and 3-cycle coatings, which serve as solid proof of the film coverage. However, we assumed that the connectivity along the MOF films was not sufficient below 5 cycles, because the slope of the $\log(R)$ vs. number of cycles graph changed — indicating a variation in resistance reduction rate per cycle (see the right panel of the manuscript, **Figs. 2e, f**). Although this is an indirect evidence, it is quite reasonable to evaluate the microstructure of the films through resistance changes, since it has been previously reported that MOFs initially grow via an island-like nucleation mechanism (*Angew. Chem. Int. Ed.* 2021, 60(49), 25758-25761).

We agree with your concerns that it is difficult to identify connectivity based on SEM images. To maximize process efficiency relative to performance, we accelerated the synthesis speed, which resulted in increased roughness of the film—making it difficult to accurately measure the thickness using techniques such as AFM. Nevertheless, EDS mapping clearly confirms the successful formation of the MOF layer (**Figs. R2–R4** in the above [R1-A3] answer).

As mentioned in the above response, extending the reaction and cleaning time could improve surface roughness by completely removing residual salts at each step. However, increasing the process time may not be the optimal choice from an engineering perspective since there is no batch-to-batch difference in the device performance. We conducted multiple batches (2–4 times) for each sensor condition, and the variation remained minimal (< 10%), indicating that this roughness does not significantly affect the gas sensing performance. In future work, we will further refine the reaction and cleaning times to enhance the film structure.

<After Correction> Results-Thickness dependent gas sensing characteristics of CuHHTP-xC sensors (page 7, line 146)

(Original) This result suggests that there were not enough reaction sites for ethanol in the initial stages of cMOF coating, which gradually increased as the number of coatings increased. As evidence, the Cu_3HHTP_2 films did not completely cover the electrode before 5 cycle coatings (**Supplementary Fig. 4**). Furthermore, the R_a of CuHHTP-xC sensors drastically decreased by 58.8% per cycle before 5 cycles, but beyond that, the rate of decrease reduced to 19.4% (**Figure 2e**).

→ This result suggests that there were not enough reaction sites for ethanol in the initial stages of cMOF coating, which gradually increased as the number of coatings increased. As

evidence, the Cu_3HHTP_2 films did not completely cover the electrode before 5 cycle coatings (Supplementary Fig. 4). Furthermore, the R_a of CuHHTP-xC sensors drastically decreased by 58.8% per cycle before 5 cycles, but beyond that, the rate of decrease reduced to 19.4% (Figure 2e). As evidence of the insufficient amount of sensing material in the early cycles, the resistance (R_a) of the CuHHTP-xC sensors sharply decreased by 58.8% per cycle during cycles 1 to 5. However, after the 5th cycle, the rate of decrease significantly slowed to 19.4% per cycle (Fig. 2e).

[R1-Q5] Line 143-Line 145, what is the difference between the in-plane and out-of-plane growth of Cu₃HHTP₂. How could the authors distinguish it according to the SEM images?

[R1-A5] Thank you very much for your valuable comment. In our original manuscript, we used the terms ‘in-plane’ and ‘out-of-plane’ with the following intended meanings:

‘Out-of-plane’ refers to the increase in Cu₃HHTP₂ film thickness.

‘In-plane’ refers to the enhanced lateral connectivity along the Cu₃HHTP₂ films.

However, we acknowledge that these terms may not have clearly conveyed the intended phenomena. To improve clarity and avoid potential misunderstandings, we have revised these expressions in the updated manuscript.

Regarding the reviewer’s point about the difficulty in confirming lateral connectivity through SEM images, we fully agree. The initial nuclei and primary particles are known to form very small, island-like MOF clusters that are spaced only a few nanometers apart (*Angew. Chem. Int. Ed.* 2021, 60(49), 25758-25761). Due to this extremely short distance, it is challenging to clearly distinguish these features using SEM imaging. We sincerely appreciate the reviewer’s insightful feedback, which helped us refine the terminology and improve the clarity of our manuscript.

<After Correction> Results-Thickness dependent gas sensing characteristics of CuHHTP-xC sensors (page 7, line 151)

(Original) As evidence, the Cu₃HHTP₂ films did not completely cover the electrode before 5 cycle coatings (**Supplementary Fig. 4**). Furthermore, the R_a of CuHHTP-xC sensors drastically decreased by 58.8% per cycle before 5 cycles, but beyond that, the rate of decrease reduced to 19.4% (**Figure 2e**). This initial rapid decrease in resistance can be attributed to both in-plane and out-of-plane growth of Cu₃HHTP₂ films, allowing the connection of Cu₃HHTP₂ nucleates in an island-type that slowly extends and merges with the surrounding nuclei³⁴. In contrast, the subsequent slower decrease at higher coating cycles (>5 times) is solely attributed to the out-of-plane thickening, which ensure the thin and uniform growth of cMOF films (**Fig. 2f**).

→ As evidence of the insufficient amount of sensing material in the early cycles, the resistance (R_a) of the CuHHTP-xC sensors sharply decreased by 58.8% per cycle during cycles 1 to 5. However, after the 5th cycle, the rate of decrease significantly slowed to 19.4% per cycle (**Fig. 2e**). Although CuHHTP films are observed to form over a wide area from the very first cycle (**Supplementary Fig. 6a**), the internal connectivity of the film may still be insufficient at this stage, as the initial nuclei and primary particles have not fully grown or merged.³⁶ Consequently, during the early cycles, the steep drop in R_a is likely due to the combined effects of thickness growth and lateral interconnection between the initially formed nuclei and/or primary particles. In contrast, at the higher coating cycles (>5 times), lateral film growth appears to reach saturation, and only vertical (thickness) growth continues—accounting for the slower rate of R_a decrease (**Fig. 2f**).

[R1-Q6] Line 145-147, is there any reference to support this statement? It is correct that connections are very important for the resistance decrease. But if the authors check the conductivity of the thin film, which excludes the effect of thickness, they will figure out that defects decrease during the LBL growth is also a very important factor. Please be careful about such statements without solid proofs.

[R1-A6] Thank you very much for your valuable comment. In the original version of the manuscript, lines 139–140 read as follows: “*In contrast, the subsequent slower decrease at higher coating cycles (>5 times) is solely attributed to the out-of-plane thickening, which ensure the thin and uniform growth of cMOF films (Fig. 2f).*”

We agree that defect control can be an important factor in reducing resistance. Indeed, some studies using ZIF-8 have shown that the layer-by-layer (LBL) method can gradually remove defect from ZIF-8 membrane (*Chem. Commun*, 2019, 55, 10146-10149).

We also acknowledge that it would be meaningful to consider whether defects are diminished as the LBL cycles proceed in our Cu₃HHTP₂ thin-film system. However, in our current experiment, the R_a values do not decrease gradually but rather exhibit a sharp change at 5 cycle coatings. This suggests that the observed behavior may not be due to a gradual defect healing process, but instead a result of the transition from disconnected and more fully connected nuclei or primary particles during LBL deposition. Furthermore, since the conductivity measurement reflects not only the defects within the crystals but also the disconnection between nuclei or primary particles (as we have answered in comment [R1-A5]), as well as grain boundary-related resistance, it is challenging to deconvolute these factors to isolate the effect of defect healing.

Nonetheless, we agree that defect healing is an important aspect to consider, and we will revise the manuscript to include this point and avoid potential misunderstandings by the readers.

<After Correction> Results-Thickness dependent gas sensing characteristics of CuHHTP-xC sensors (page 8, line 160)

(Original) In contrast, the subsequent slower decrease at higher coating cycles (>5 times) is solely attributed to the out-of-plane thickening, which ensure the thin and uniform growth of cMOF films (**Fig. 2f**).

→ Consequently, during the early cycles, the steep drop in R_a is likely due to the combined effects of thickness growth and lateral interconnection between the initially formed nuclei and/or primary particles. In contrast, at the higher coating cycles (>5 times), lateral film growth appears to reach saturation, and only vertical (thickness) growth continues—accounting for the slower rate of R_a decrease (**Fig. 2f**). Furthermore, the gradual reduction in resistance could also be attributed to ongoing defect healing within the MOF throughout the entire LBL cycling process. For instance, Ma et al. demonstrated that defects of ZIF-8 membranes during the early LBL cycles are eliminated after 10 continuous cycles of LBL coating³⁷.

[R1-Q7] Line 170-171, why do cMOFs with a light source enhance the repeatability? Could you explain more about this statement (for instance, the improved recovery)?

[R1-A7] Thank you very much for your comment. In the original version of the manuscript, lines 139–140 read as follows: “*Integrating cMOFs sensors with a light source, though, provides a viable pathway to enhance their repeatability.*”

At room temperature, the thermal energy is often insufficient to overcome the activation energy barrier required for complete desorption of certain strongly adsorbed gases (such as NO₂ and H₂S), which limits the recovery of the sensing signal. This limitation is particularly significant in many conducting MOFs (cMOFs) [*Nat. Commun.* 2021, 12, 4294; *Adv. Mater.* 2022, 34(12), 2107696], restricting their practical applicability. In 2021, we demonstrated that light activation can enhance the NO₂ desorption process in cMOFs by generating additional charge carriers via illumination with a bulk LED [*ACS Cent. Sci.* 2021, 7(7), 1176–1182]. As a follow-up, we propose further miniaturizing the sensor system by integrating a μ LED to facilitate on-chip light activation. We have added these references in the revised manuscript to include more detailed information to clarify this point.

<After Correction> Results-Thickness dependent gas sensing characteristics of CuHHTP-xC sensors (page 9, line 191)

(Original) This limitation highlights a significant challenge in achieving repeatable use for many room-temperature gas sensors. Integrating cMOFs sensors with a light source, though, provides a viable pathway to enhance their repeatability²⁰.

→ This limitation highlights a significant challenge in achieving repeatable use for many room-temperature gas sensors.^{40,41} Integrating cMOF sensors with a light source, though, generates additional photogenerated charges that drive the reverse reaction, facilitating the desorption of adsorptive gases and thereby enhancing their reversibility to realize repeatable response and recovery²⁰.

[R1-Q8] Line 229- 230, did you conduct any characterizations for the film to support this statement? Ni-HHTP or Co-HHTP is very difficult to form a crystalline thin film by such methods, it is quite difficult to tell how they connect if they are just amorphous coordination polymers without periodic structures.

[R1-A8] Thank you very much for your comment. In the original version of the manuscript, lines 229–230 read as follows: “During this process, the two 2D MOF layers are presumed to form local alloy phases, where Co and Cu coordinate through a single HHTP ligand in-plane, or the Co_3HHTP_2 and Cu_3HHTP_2 layers stack out-of-plane.”

Our hypothesis is based on two key points: It has already been proven that triphenylene-based ligands can form alloy structures when two different metal ions are mixed (*J. Am. Chem. Soc.*, 2020, 142 (28), 12367–12373). Additionally, it has been revealed that even when the two metals are mixed, the conductivity remains continuous through the ligand. Our Ni_3HHTP_2 and Co_3HHTP_2 overlayer coating process was conducted immediately and continuously after the formation of the Cu_3HHTP_2 -based films, while the reaction was still incomplete. Therefore, we argued that a similar local alloy was formed, and based on these factors, we stated that “...are presumed to form a local alloy phase.”

However, there is no solid evidence to support these statements, as Ni_3HHTP_2 and Co_3HHTP_2 layers are too thin to characterize their crystallinity. We only confirmed that both layers exist on the surface of the Cu_3HHTP_2 films (**Supplementary Figs. 9 and 10**).

Supplementary Fig. 9 | a Cross-section view SEM image of NiHHTP-7C/CuHHTP-15C on Si/SiO₂ substrate. b–e Energy-dispersive X-ray spectroscopy (EDS) mapping of NiHHTP-7C/CuHHTP-15C on Si/SiO₂ substrate; (b) C element, (c) O element, (d) Ni element, and (e) Cu element. The substrate is tilted by 3° to observe the presence of both the Ni element in the overlayers and the Cu element in the base layer.

Supplementary Fig. 10 | a Cross-section view SEM image of CoHHTP-7C/CuHHTP-15C on Si/SiO₂ substrate. b–e Energy-dispersive X-ray spectroscopy (EDS) mapping of CoHHTP-7C/CuHHTP-15C on Si/SiO₂ substrate; (b) C element, (c) O element, (d) Co element, and (e) Cu element. The substrate is tilted by 3° to observe the presence of both the Co element in the overlayers and the Cu element in the base layer.

In this system, Cu₃HHTP₂ conducting films were already formed after 5 cycles. Any further coating has a minor effect on conductivity but rather contributes to the gas reaction through filtration or catalytic properties (Fig. 2e). Since our focus is on the catalytic properties of Co₃HHTP₂ and Ni₃HHTP₂, the crystallinity of these films was not a critical factor.

Therefore, we removed and revised the corresponding statement in the manuscript to avoid any potential controversy.

<After Correction> Results-Overlayer coating of M₃HHTP₂ (M = Ni, Co) on CuHHTP-5C films (page 12, line 248)

(Original) For methyl-oxygen interactions, the binding energies were relatively low across all MOFs, ranging from -0.33 eV to -0.37 eV, presumably due to the reduced influence of metal sites. Thus, this suggests that TMA gas exhibits stronger adsorption on Co₃HHTP₂. The Co₃HHTP₂ overlayers are epitaxially stacked onto the Cu₃HHTP₂ base layers through continuous immersion cycles of ligand and metal solutions. During this process, the two 2D MOF layers are presumed to form local alloy phases, where Co and Cu coordinate through a single HHTP ligand in-plane, or the Co₃HHTP₂ and Cu₃HHTP₂ layers stack out-of-plane. As a result, redox reactions occurring at the Co₃HHTP₂ layer can influence the overall resistance of the film and, in the case of TMA, enhance sensitivity.

→ For methyl-oxygen interactions, the binding energies were relatively low across all MOFs, ranging from -0.33 eV to -0.37 eV, presumably due to the reduced influence of metal sites.

In addition, a comparison of the Cu²⁺/Cu⁺ ratio between CuHHTP-5C (Supplementary Fig. 3) and CoHHTP-7C/CuHHTP-5C (Supplementary Fig. 12) showed that Cu²⁺ was reduced to Cu⁺ after the Co₃HHTP₂ overlayer coating. This suggests that the electrical signal variation in the Co₃HHTP₂ layers induced by TMA adsorption can be transferred to the bottom Cu₃HHTP₂ layers, and changes the overall resistance of the film and enhancing its sensitivity.

~~Thus, this suggests that TMA gas exhibits stronger adsorption on Co₃HHTP₂. The Co₃HHTP₂ overlayers are epitaxially stacked onto the Cu₃HHTP₂ base layers through continuous immersion cycles of ligand and metal solutions. During this process, the two 2D MOF layers are presumed to form local alloy phases, where Co and Cu coordinate through a single HHTP ligand in-plane, or the Co₃HHTP₂ and Cu₃HHTP₂ layers stack out-of-plane. As a result, redox reactions occurring at the Co₃HHTP₂ layer can influence the overall resistance of the film and, in the case of TMA, enhance sensitivity.~~

[R1-Q9] Line 259, please add the full name of IDE?

[R1-A9] Thank you for pointing this out. The full name of IDE is 'Interdigitated Electrode'. In the original manuscript, the abbreviation was used without the full name, and we have now revised it accordingly.

<After Correction> Results-Integration of cMOFs and μ LP (page 14, line 292)

(Original) The μ LPs were fabricated using micro-electro-mechanical system (MEMS) processes, insulated with a double-layered SiO₂, and patterned with IDE.

→ The μ LPs were fabricated using micro-electro-mechanical system (MEMS) processes, insulated with a double-layered SiO₂, and patterned with **interdigitated electrode (IDE)**.

[R1-Q10] Line 292-294, as mentioned in comment 1, for a reader-friendly version, the authors are recommended to clarify the following question: will the light activation affect the NO₂ or NH₃ gas adsorption and desorption on the film? You'd better take this influence into consideration when you analyze why light activation enhanced reversibility. Why light activation enhanced reversibility for NO₂ and NH₃ gas needs deep analysis and discussion.

[R1-A10] Thank you very much for your comment. We realized that we had missed providing a detailed sequence of the photoactivation and recovery process. For NO₂ and NH₃, due to their strong adsorptive properties (*Adv. Sci.*, 2019, 6(21), 1900250. *Angew. Chem. Int. Ed.*, 2019, 131(42), 15057-15061) the forward reactions in Equations 1 and 2 proceed favorably.

In this case, if the number of electrons or holes increases, the reverse reaction can be enhanced, thereby accelerating the desorption of the adsorbed NH₃ and NO₂ gases.

In this study, we utilized blue μ LED (2.73 eV) or UV μ LED (3.14 eV), which provide energy to excite the π - π^* transition energy gap (~2.90 eV) of the HHTP-based MOF, generating photoelectrons and thereby facilitating the recovery of adsorbed gases. Notably, this approach enabled the reversible recovery of NO₂, which is typically considered irreversible. We have identified the lack of detailed explanation in the main text and have now supplemented it with the following additional description.

Additionally, **Fig. R8** in [R3-A1] shows that our own DFT calculations also confirm stronger adsorption energies of NH₃ and NO₂ on Cu₃HHTP₂.

<After Correction> Results-Integration of cMOFs and μ LP (page 15, line 320)

(Original) This is presumably because the high-energy UV light not only increases the reaction sites for the neutral methyl group but also accelerates the desorption of basic amine sites of TMA gases. In contrast, when sensing basic NH₃ and acidic NO₂ using CuHHTP-5C sensors, the reverse reaction became more dominant under light activation due to their excessive adsorption, even in dark conditions as shown in Eqs. (1) and (2) (**Supplementary Fig. 17**).

Although the correlation between light intensity and the response is not clearly demonstrated, the sensitivity decreases when the sensor is exposed to light. For NO_2 , light activation is beneficial due to its strong adsorption characteristics, which significantly improve the recovery rate, thereby enhancing reversibility (**Fig. 5f**)

→ This is presumably because the high-energy UV light not only increases the reaction sites for the neutral methyl group but also accelerates the desorption of basic amine sites of TMA gases. In contrast, when sensing basic NH_3 and acidic NO_2 using CuHHTP-5C sensors, the forward reaction is strongly favored, as shown in Eqs. (1) and (2) (**Supplementary Fig. 20**). In particular, NO_2^- adsorbates are irreversibly bound to the MOF surfaces. At this stage, light illumination can generate electron-hole pairs, thereby accelerating the reverse reaction and promoting the desorption of adsorbates.

Although hindrance of the forward reaction can limit gas sensitivity, light activation is beneficial in the case of NO_2 , significantly improving the recovery rate and thereby enhancing reversibility (**Fig. 5f**).

Reviewer #2 (Remarks to the Author):

In this manuscript Park et al. sophisticatedly designed and prepared conductive MOF thin film arrays on micro-LEDs for gas sensing. Comprehensive characterizations and investigations were conducted on this system to demonstrate the work mechanism. A so-called e-nose technology was developed for gas detection. Basically, I think it holds the novelty and significance for *Nat. Commun.* The manuscript is well written. It could be acceptable if the following issues could be addressed:

We sincerely appreciate Reviewer #2 for the insightful comments and valuable suggestions. In response to the reviewer's feedback, we have carefully addressed each point in detail. We have created correction boxes to clearly show the changes made in the revised manuscript, with both the original (before) and revised (after) texts provided for each modification.

*Please note that the page and line numbers indicated in each correction box correspond to the highlighted version of the revised manuscript, where all changes are marked for clarity.

[R2-Q1] More discussion should be given to clarify why the four specific gases (EtOH, TMA, NH₃ and NO₂) were chosen. What kind of application scenarios is this gas sensing technology used for? What is the challenge?

[R2-A1] We appreciate the reviewer's comment. The selection of the four target gases—ethanol (EtOH), trimethylamine (TMA), ammonia (NH₃), and nitrogen dioxide (NO₂)—was based on their distinct chemical properties, allowing us to systematically evaluate the sensor's performance. These gases were chosen to represent a balanced combination of oxidizing vs. basic and non-adsorptive vs. adsorptive interactions with the cMOF film. This design enabled a comprehensive investigation into how the film composition and photoactivation conditions affect sensitivity, recovery, and selectivity, leading to an optimized sensing platform.

The selected four target gases are also closely related to indoor air quality and are gases that are 'plausibly present' in our daily lives. EtOH and TMA are commonly used as indicators in food spoilage detection, where EtOH is released during fermentation, and TMA is a key marker of fish freshness. NH₃ is widely studied in environmental monitoring and medical diagnostics, particularly in exhaled breath analysis for metabolic and respiratory conditions. NO₂ is an important air pollutant associated with vehicle emissions and industrial activities.

Key challenges in real-world applications include maintaining selectivity in complex gas mixtures, ensuring long-term stability for repeated use, and improving recovery time to enable real-time monitoring. Our ongoing research is further exploring these challenges, particularly in the development of cMOF-based sensors capable of selectively detecting target gases in mixed-gas environments. Although this is still ongoing research, we have included feasibility results demonstrating that the cMOF chemiresistor array can selectively distinguish mixed gases (please see our response to **[R2-Q4]** and **Fig. R6**).

[R2-Q2] For the c-MOF, why this MHHTP was selected? During the fabrication of MOF overlayers, it is quite possible that the lattices will merge with each other to form a mixed metal layer, as shown in **Supplementary Figs. 8d-e**. Is there any effect on the sensing performances? Please clarify this.

[R2-A2] We thank the reviewer for their comments. **Supplementary Fig. 8** is now **Fig. 9** after revision.

MHHTP was selected for this study as it is one of the most extensively studied and well-characterized systems, and it exhibits robust gas sensing characteristics. Furthermore, for practical applications, the commercial availability of the HHTP ligand is an important consideration.

Within the MHHTP series, Cu_3HHTP_2 was chosen as the base conducting layer due to its superior electrical conductivity compared to its Ni- and Co-based counterparts. In contrast, Ni_3HHTP_2 and Co_3HHTP_2 MOFs were used to form catalytic overlayers that serve as gas adsorption layers for the different reactive adsorbates. The conductivity changes occurring in these overlayers are efficiently transferred to the underlying Cu_3HHTP_2 base layer, owing to the electrically conductive nature of cMOFs. Related discussions can be found in the ‘Overlayer coating of M_3HHTP_2 (M = Ni, Co) on CuHHTP-5C films’ in the Results section.

In addition, the formation of overlayers is a widely used approach in oxide-based gas sensors to enhance sensitivity and modulate selectivity (*Nat. Commun.* 2023, 14, 233, *Adv. Sci.* 2021, 8(6), 2004078). Therefore, adopting a similar strategy for MOF-based chemiresistive sensors is well justified.

As noted by Reviewer #2, we presumed that the heterometal overlayers merge with the base Cu_3HHTP_2 layers to form an alloy-like system, consistent with previously reported references (*J. Am. Chem. Soc.* 2020, 142, 28, 12367–12373). However, in our experiments, it was challenging to directly confirm this due to the thin and rough nature of the film surfaces. The surface agglomeration observed is likely a result of the rapid layer formation process, which involves immersion of the substrate in the ligand solution for 2 minutes, followed by a 1-minute exposure to the metal solution, and a brief cleaning step lasting only tens of seconds (The agglomerated particles are not impurities but confirmed as cMOFs, as shown in **Supplementary Fig. 6**).

Extending the reaction and cleaning durations could improve surface morphology by more effectively removing residual salts at each step. However, from an engineering standpoint, if there is no significant batch-to-batch variation in device performance, prolonging the process may not be justified. We performed 2–4 independent fabrication batches for each sensor condition, and the variation in performance remained minimal (<10%), suggesting that the surface roughness does not substantially affect gas sensing characteristics (**Supplementary Fig. 5**).

Nonetheless, we acknowledge the importance of optimizing film uniformity and, in future work, we plan to further refine the reaction and cleaning protocols to improve the structural quality of the films.

[R2-Q3] The deep-learning is difficult to understand. More introduction and discussion should be provided. After the deep-learning, what have been achieved as great improvements?

[R2-A3] Thank you for your comment. The deep learning structure used in this study is described in **Supplementary Note 3**. However, we acknowledge that some readers may not be familiar with deep learning, and we agree that providing a more detailed and intuitive explanation of the convolutional neural network (CNN) algorithm used in this study would improve understanding. CNN is widely used for processing image and video data, and it has also been actively applied to sensor systems in recent years. Unlike deep neural networks (DNNs), which flatten input data and lose spatial information, CNNs preserve spatial relationships within the data, such as maintaining the correlation between a pixel and its neighboring pixels in a 2D image, allowing for more effective feature extraction.

In our sensor system, the input data can be structured similarly to 2D data. Our cMOF-based chemiresistive sensor generates a temporal resistance signal, where the x -axis represents time and the y -axis represents the signal response, forming a $1 \times (\text{time})$ matrix. By concatenating signals from all four sensors along the vertical axis, ensuring time alignment since all sensors were operated simultaneously, we obtained a $4 \times (\text{time})$ matrix. To enhance feature extraction, we applied a sliding time window before CNN processing, which helps address the low selectivity issue that arises when relying only on response magnitude (R_g/R_a) to distinguish different gases. More details on this approach can be found in **Fig. R5**, which is from our previous study [*ACS Nano* 2023, 17, 1, 539–551]. For CNN training, we used the sliding time window size of 60 s, resulting in an input data size of 4×60 . A 6-channel convolutional filter (kernel) of size 4×30 was applied, followed by fully connected layers. The final output consisted of six nodes, five of which were used for gas classification, where each node provided the probability for one of the five gas types (air, EtOH, TMA, NH_3 , and NO_2). The remaining node was used for gas concentration prediction. During CNN training, the model was optimized to minimize training loss, ensuring that the overall accuracy of bimodal detection, which includes gas classification and concentration prediction, was maximized. Hyperparameters like epochs, learning rate, batch size, stride, number of filters, optimizer type, were fine-tuned accordingly to achieve the best performance.

The deep learning approach in this study has led to two major improvements.

First, it effectively addresses the low selectivity issue inherent in chemiresistive gas sensors, enabling both gas type classification and concentration prediction. Traditional chemiresistive sensors often struggle with distinguishing gases due to overlapping response patterns, but by utilizing CNN-based pattern recognition, our system achieves highly accurate identification and quantification of target gases.

The second major improvement is the significant reduction in prediction time. Although photoactive gas sensors, including our $\mu\text{LED} + \text{cMOF}$ system, offer advantages such as low power consumption and room-temperature operation, they generally suffer from slow response and recovery times. CNN, as a supervised learning method, allows the model to predict gas type and concentration even before the response reaches full saturation. By analyzing early-stage response characteristics, such as the initial slope and shape of the signal, the model can

make rapid and reliable predictions. According to **Supplementary Fig. 20**, although the sensor's response (T_{90}) and recovery (T_{10}) times span tens of minutes, the CNN-based deep learning approach enables gas type and concentration to be predicted within two minutes during both response and recovery phases, representing a substantial improvement in real-time gas detection (Please also refer to [R3-A9]).

In response to the reviewer's valuable comment, we have substantially revised **Supplementary Note 3** and significantly enhanced the level of detail.

Fig. R5. a–c. The difference between the CNN and other ML algorithms such as PCA, KNN, SVM, etc. Training process and output results of (a) traditional ML and (b) CNN method. (c) Table comparing CNN and other ML algorithms [ACS Nano 2023, 17, 1, 539–551].

<After Correction> Supporting Information - Supplementary Note 3

→ In this study, the e-nose system is composed of a 4-cMOF sensor array and a convolutional neural network (CNN)-based deep learning algorithm. Each individual cMOF sensor

generates a temporal resistance signal, where the x -axis represents time and the y -axis represents the signal response, forming a $1 \times (\text{time})$ matrix. Since all sensors were operated simultaneously, the signals were concatenated along the vertical axis to construct a $4 \times (\text{time})$ matrix. The collected signals were then normalized to R_g/R_a and structured into a 4 (number of sensors) \times 7000 seconds (gas test duration) matrix. The detailed CNN architecture is shown in **Supplementary Fig. 27**.

CNNs are particularly well-suited for tasks that require the extraction of spatially or temporally correlated features from structured data. Unlike fully connected deep neural networks (DNNs) that flatten input matrices and lose local correlation information, CNNs are able to preserve and exploit the spatial (or temporal) structure of input data, making them highly effective for analyzing time-dependent signals. In our system, the transient signals of the sensor over time exhibit uniquely different profiles for each gas and carry critical information. CNNs are advantageous in this context because they can recognize these differences, which are difficult to capture using only static features such as the response magnitude (e.g., R_g/R_a at the maximum point of the gas response).

To preprocess the data, a 60-second sliding time window moving at 1-second intervals was applied, converting the original 4×7000 matrix into 6940 matrices of size 4×60 . This sliding window approach enables the model to not only analyze steady-state response magnitudes (R_g/R_a) but also to capture unique dynamic features such as the initial slope, rise shape, and transient fluctuations specific to each gas. As a result, the system's ability to distinguish between gases is significantly enhanced beyond simple magnitude-based classification.

A 2D convolutional layer with a 4×30 filter size and 6 output channels was applied, using a stride of 2 seconds. Each convolutional layer was followed by batch normalization (BN) and a leaky rectified linear unit (leaky-ReLU) activation function to improve training stability and generalization. After convolution, the extracted features were passed through three fully connected (FC) layers consisting of 32, 16, and 8 nodes, respectively, each also employing batch normalization and leaky-ReLU activation.

The final output structure was designed for dual tasks: gas classification and concentration prediction. The classification output consisted of 5 nodes corresponding to air, EtOH, TMA, NH_3 , and NO_2 classes, using a softmax function to predict the gas label with the highest probability. The regression head included 1 node to predict gas concentration. For improved robustness and accuracy of concentration predictions, the TRIMMEAN function was applied as post-processing within the 60-second time window, excluding the top and bottom 20% of values to minimize the effect of outlier fluctuations.

The CNN model was trained by minimizing a combined loss function, defined as:

$$L_{\text{total}} = w \cdot L_{\text{cross-entropy}} + (1 - w) \cdot L_{\text{MSE}} (w = \text{weight}),$$

where $L_{\text{cross-entropy}}$ represents the categorical cross-entropy loss for classification, L_{MSE} is the mean squared error for regression, and w is the weighting factor between the two tasks. The Adam optimizer was used for training with a learning rate $\eta = 10^{-4}$. Through hyperparameter tuning, the final values were set as $w = 0.05$ and epochs = 2900. Other tuned hyperparameters included the batch size, stride, number of filters, and kernel size to optimize model performance.

Importantly, the CNN approach led to two major improvements compared to using only the sensor responses and traditional machine learning techniques:

(1) It significantly enhanced gas selectivity by leveraging time-dependent dynamic features (transient response) rather than relying solely on response magnitude (R_g/R_a).

(2) It dramatically reduced the prediction latency, enabling gas type and concentration to be identified within about one minute. Including the initial sliding window accumulation, the total decision time stayed under two minutes, which is still substantially faster than the intrinsic response and recovery times of the cMOF sensors (typically tens of minutes, as shown in **Supplementary Fig. 20**).

Overall, the CNN-based e-nose system presented in this study provides highly selective, fast, and accurate real-time gas prediction.

[R2-Q4] In practical applications, gas sensing is usually used in complicated gas mixture. The potential of such applications should be conducted.

[R2-A4] We appreciate your insightful comment. In this study, our primary focus was on evaluating the sensor's fundamental characteristics under controlled single-gas conditions. The scope of this work was to meticulously design and optimize the cMOF-based sensor by precisely tuning the film thickness and integrating Ni₃HHTP₂ and Co₃HHTP₂ overlayers onto the Cu₃HHTP₂ base layer. The selected four target gases were carefully chosen to represent different chemical interactions—oxidizing vs. basic, and non-adsorptive vs. adsorptive—allowing us to systematically investigate their sensitivities and recovery behaviors. Through this approach, we explored the optimal film composition and photoactivation conditions to enhance sensor performance and provided an in-depth analysis and discussion on why the selected film configuration was the most effective. As the reviewer pointed out, evaluating the sensor's performance in mixed-gas environments is a crucial step toward practical applications. However, selectively detecting specific gases in complex gas mixtures using the cMOF + μ LED sensor is part of our ongoing research. As a proof-of-concept study, we conducted preliminary experiments with EtOH and NH₃ gas mixtures. Specifically, we tested three cases: (1) 50 ppm EtOH alone, (2) 5 ppm NH₃ alone, and a (3) mixture of both gases using the CuHHTP-5C sensor at dark (LED OFF) and CuHHTP-15C sensor at UV L1. As shown in **Fig. R6**, the CuHHTP-5C sensor, which has a relatively thin film, exhibited higher sensitivity to NH₃ compared to CuHHTP-15C. In contrast, the trend was reversed for EtOH, where the thicker CuHHTP-15C film showed a higher response. These results further confirm that the optimal film thickness for maximizing sensitivity depends on the chemical properties of the target gas. Moreover, the mixed-gas test results indicate that the sensor response is a superposition of the individual responses from each gas in the mixture. This finding highlights the potential for selective detection of mixed gases by leveraging the gas-specific affinity of different cMOF film configurations and CNN-based signal processing. As mentioned earlier, we are currently conducting more in-depth studies on gas mixtures, and therefore, we prefer to include this discussion in the response letter rather than the main manuscript.

Fig. R6. a–c Gas sensing performance of CuHHTP films under mixed-gas exposure of EtOH and NH₃. (a) Response curve of the CuHHTP-5C sensor under dark conditions when exposed to a mixture of 50 ppm EtOH and 5 ppm NH₃. (b) Response curve of the CuHHTP-15C sensor under UV L1 illumination under the same mixed-gas condition. (c) Bar graph comparing the response magnitudes of the cMOF sensors in (a) and (b), illustrating the feasibility of selective detection of mixed gas.

[R2-Q5] Except for the NO₂ gas, other gases, as they stated, are reversible. How is about the recyclability? I suggested collecting the experimental data, showing the sensing performances after tens of times.

[R2-A5] Thank you for the valuable comment. To evaluate the recyclability of our cMOF-based sensors, we conducted 20 repeated cyclic tests using the optimal sensor and photoactivation conditions for each gas, as presented in the main manuscript: 50 ppm EtOH, 50 ppm TMA, and 5 ppm NH₃. As shown in the **Fig. R7**, the sensors developed in this study exhibit excellent recyclability, maintaining stable sensing performance over repeated cycles. These results have been added as **Supplementary Fig. 22**.

Fig. R7. a–c Repeatability test of optimized cMOF sensors under cyclic gas exposures. (a) CuHHTP-15C (UV L1) sensor to 50 ppm EtOH. (b) CoHHTP-7C/CuHHTP-5C (Blue L2) sensor to 50 ppm TMA. (c) CuHHTP-5C (Dark) sensor to 5 ppm NH₃.

<After Correction> Results- Integration of cMOFs and μ LP (page 16, line 349)

(Original) With their optimal sensitivity, all sensors were reversible, enabling repeated and continuous gas detection with minimum power consumptions.

→ With their optimal sensitivity, all sensors were reversible, enabling repeated and continuous gas detection with minimum power consumptions. **Cycle tests under these optimized conditions (Supplementary Fig. 22) confirmed good repeatability, with μ LED photoactivation aiding recovery of the cMOF sensors.**

Supplementary Fig. 22 | a–d Repeatability test of optimized cMOF sensors under cyclic gas exposures. (a) CuHHTP-5C (Blue L2) sensor to 2 ppm NO₂ over 10 cycles. (b) CuHHTP-15C (UV L1) sensor to 50 ppm EtOH over 20 cycles. (c) CoHHTP-7C/CuHHTP-5C (Blue L2) sensor to 50 ppm TMA over 20 cycles. (d) CuHHTP-5C (Dark) sensor to 5 ppm NH₃ over 20 cycles.

[R2-Q6] Some typos: “It presumably resulting in intermediate response trends between non-adsorptive gas and highly adsorptive gases (e.g., NH₃).”

[R2-A6] Thank you for pointing this out. We have identified the grammatical issue and revised the sentence accordingly in the manuscript.

<After Correction> Results-Thickness dependent gas sensing characteristics of CuHHTP-xC sensors (page 9, line 184)

(Original) It presumably resulting in intermediate response trends between non-adsorptive gases (e.g., EtOH) and highly adsorptive gases (e.g., NH₃).

→ It presumably ~~resulting~~ **results** in intermediate response trends between non-adsorptive gases (**such as** EtOH) and highly adsorptive gases (**such as** NH₃).

Reviewer #3 (Remarks to the Author):

This study presents a novel integration of photoactivated conductive metal-organic framework (cMOF) thin films with micro-LEDs (μ LEDs) for chemiresistive gas sensing, achieving ultra-low power consumption ($587 \mu\text{W}$) and high classification accuracy (99.8%) using deep learning. The systematic optimization of cMOF film thickness, overlayer composition, and μ LED illumination parameters demonstrates a compelling approach to address challenges in room-temperature gas sensing, such as reversibility and selectivity. The combination of experimental data, DFT calculations, and CNN-based pattern recognition provides a robust framework for advancing e-nose technology. However, several methodological details, data interpretations, and technical validations require clarification to strengthen the claims and ensure reproducibility.

We sincerely appreciate the Reviewer #3's thoughtful and encouraging feedback. We are pleased that the reviewer recognized the novelty of our photoactivated cMOF gas sensor, which is based on an ultra-low power μ LED and supports the development of an e-nose through deep learning. We also thank the reviewer for prompting us to incorporate more fundamental and in-depth discussions during the revision process. In response to the reviewer's concerns, we have thoroughly revised the manuscript to clarify methodological details and provide additional validation results where necessary. We believe these revisions have significantly improved the clarity, rigor, and reproducibility of our work. We have created correction boxes to clearly show the changes made in the revised manuscript, with both the original (before) and revised (after) texts provided for each modification.

*Please note that the page and line numbers indicated in each correction box correspond to the highlighted version of the revised manuscript, where all changes are marked for clarity.

[R3-Q1] In **Fig. 2d**, the response trends for NH_3 and NO_2 decrease with increasing CuHHTP- x C cycles, while EtOH responses saturate. What is the quantitative relationship between film thickness (**Supplementary Fig. 4**) and gas adsorption kinetics? Provide diffusion coefficients or adsorption energy calculations to support the "gas filtering effect" hypothesis.

[R3-A1] We appreciate your insightful comment. To our knowledge, highly adsorptive gases tend to exhibit low diffusivity. For example, Gang et al. (<https://doi.org/10.1002/anie.201907772>) demonstrated that a uniformly deposited Cu-TCPP overlayer on a Cu-HHTP sensing film acted as a molecular sieving layer by selectively adsorbing NH_3 , which has a high affinity for the Cu sites of the Cu-TCPP layers. This resulted in reduced responses to adsorptive NH_3 gases due to the filtering effect, while enhancing selectivity for benzene, a less adsorptive gas.

A similar effect was observed in our study. To support our hypothesis, we calculated the binding energies of EtOH, NH_3 , and NO_2 with Cu-HHTP MOFs. NH_3 and NO_2 exhibited high binding energies (adsorption energies) of -0.51 eV and -0.52 eV , respectively, while EtOH showed a significantly lower binding energy of -0.31 eV (**Fig. R8**).

For NH_3 and NO_2 gases, as the thickness of the Cu-HHTP layers increases, the upper part of

the Cu-HHTP layers acts as a filtering layer that limits the diffusion of NH_3 and NO_2 into the bottom Cu-HHTP layers. For reference, due to the bottom electrode design, diffusion to the lower gas-sensing material is critical for changing the device resistance. However, in the case of EtOH gas, due to its lower adsorption energy and chemically neutral properties compared to NH_3 and NO_2 , the filtering effect does not occur, and the sensing response is maintained even after thick coating of the Cu-HHTP layers.

Fig. R8. DFT-calculated adsorption energies of $\text{Cu}_3(\text{HHTP})_2$ with EtOH, NH_3 , and NO_2 molecules.

[R3-Q2] On Page 7, Lines 148-153, The irreversible recovery of NO₂ (**Supplementary Fig. 7**) is attributed to strong adsorption. However, the mechanism of light-activated reversibility (**Fig. 5f**) is not quantitatively compared to thermal desorption. Include temperature-dependent recovery experiments under dark/light conditions to isolate photochemical vs. thermal effects.

[R3-A2] Thank you very much for your insightful comment. First, we would like to clarify that all experiments in this study were conducted at room temperature (RT, ~20 °C) to eliminate any potential influence of thermal activation on the gas sensing behavior of the cMOF. This is particularly important because cMOFs are thermally sensitive and may degrade when activated at temperatures above ~150 °C. Therefore, operating the sensor at RT using an ultra-low power μLED not only ensures material stability but also provides a highly energy-efficient sensing mechanism, making it well-suited for practical applications.

In addition, as shown in **Supplementary Figs. 24** and **25**, we confirmed that μLED illumination does not cause significant photothermal heating of the cMOF films, thereby isolating the photoactivation effect from thermal interference. As the reviewer suggested, we further performed comparative gas sensing experiments under different activation conditions (**Fig. R9**). Specifically, the cMOF + μLED sensor was placed in a chamber heated to 70 °C, and measurements were taken under the following conditions: (1) heating + photoactivation, (2) heating only, (3) photoactivation only, and (4) dark at room temperature. The recovery and sensitivity followed the trend: photoactivation + heating > photoactivation only > heating only > no activation at all. These results suggest that while thermal desorption at elevated temperatures does contribute to recovery, μLED-based photoactivation is a more practical and energy-efficient method that avoids thermal degradation of the cMOF.

Because **Fig. R9** provides important evidence supporting the effectiveness and feasibility of using photoactivated cMOF sensors with integrated μLEDs, we have added it as **Supplementary Fig. 21** and included a corresponding explanation in the manuscript.

Fig. R9. Comparison of recovery levels of the CuHHTP-5C sensor after exposure to 2 ppm NO₂ under four different conditions. The sensor was tested under (1) room temperature in the dark, (2) 70 °C in the dark, (3) room temperature with blue μLED light activation (L2), and (4) 70 °C with blue μLED light activation (L2), to evaluate the effects of thermal and photo activation on cMOF sensor's recovery.

<After Correction> Results- Integration of cMOFs and μ LP (page 16, line 329)

(Original) Although the correlation between light intensity and the response is not clearly demonstrated, the sensitivity decreases when the sensor is exposed to light. For NO_2 , light activation is beneficial due to its strong adsorption characteristics, which significantly improve the recovery rate, thereby enhancing reversibility (Fig. 5f). Considering that most MOF chemiresistors suffer from irreversibility to NO_2 ⁴², their practicality was improved through a high recovery rate. In contrast, while NH_3 also shows slightly improved recovery with light activation, the corresponding decrease in sensitivity outweighs the benefit of faster recovery, leading to the conclusion that operating in dark conditions is optimal for NH_3 detection.

→ Although hindrance of the forward reaction can limit gas sensitivity, light activation is beneficial in the case of NO_2 , significantly improving the recovery rate and thereby enhancing reversibility (Fig. 5f). Considering that most MOF chemiresistors suffer from irreversibility to NO_2 ⁴⁸, their practicality was improved through a high recovery rate. Comparison under thermal and photoactivation conditions confirmed that photoactivation more effectively promotes NO_2 recovery of the cMOF sensor, demonstrating the superiority of the μ LED-integrated cMOF sensor (Supplementary Fig. 21). In contrast, while NH_3 also shows slightly improved recovery with light activation, the corresponding decrease in sensitivity outweighs the benefit of faster recovery, leading to the conclusion that operating in dark conditions is optimal for NH_3 detection.

Supplementary Fig. 21 | Comparative study of the recovery behavior of the CuHHTP-5C sensor toward 2 ppm NO_2 under thermal and photoactivation conditions. The results demonstrate that photoactivation using the blue μ LED (L2) is more effective in promoting recovery than thermal heating alone.

[R3-Q3] In Fig. 3b, CoHHTP-7C/CuHHTP-5C shows a 73.9% increase in TMA response. Clarify whether this enhancement is solely due to Co's binding energy (Fig. 4) or influenced by charge transfer between Co/Cu layers. Provide XPS or UPS data to confirm interfacial electronic interactions.

[R3-A3] We appreciate your insightful comment. Both effects play a crucial role in enhancing TMA responses.

First, from the XPS results, we observe that the $\text{Cu}^{2+}/\text{Cu}^+$ ratio in pristine Cu_3HHTP_2 layers (Fig. R10a) is significantly higher than that in $\text{Co}_3\text{HHTP}_2/\text{Cu}_3\text{HHTP}_2$ layers (Fig. R10b). This indicates that the Cu^{2+} metal is reduced to Cu^+ by interacting with Co_3HHTP_2 layers, confirming that both layers are electrically interacting with each other.

Therefore, higher affinity of TMA with Co sites increases the number of gas reactions, and their charge exchanges transfers to the Cu_3HHTP_2 layers, resulting in a significant chemiresistive variation.

Fig. R10. Cu 2p XPS spectra of (a) CuHHTP-5C and (b) CoHHTP-7C/CuHHTP-5C. Fig. R10a is identical to Supplementary Fig. 3b and Fig. R10b is identical to Supplementary Fig. 12c. A notable decrease in the $\text{Cu}^{2+}/\text{Cu}^+$ ratio is observed in (b), indicating that the application of the Co overlayer leads to a partial reduction of Cu^{2+} to Cu^+ . This reduction confirms that the Cu_3HHTP_2 and Co_3HHTP_2 layers are electronically interacting at the interface.

<After Correction> Results-Overlayer coating of $M_3\text{HHTP}_2$ ($M = \text{Ni}, \text{Co}$) on CuHHTP-5C films (page 12, line 248)

(Original) For methyl-oxygen interactions, the binding energies were relatively low across all MOFs, ranging from -0.33 eV to -0.37 eV, presumably due to the reduced influence of metal sites. Thus, this suggests that TMA gas exhibits stronger adsorption on Co_3HHTP_2 .

→ For methyl-oxygen interactions, the binding energies were relatively low across all MOFs, ranging from -0.33 eV to -0.37 eV, presumably due to the reduced influence of metal sites.

In addition, a comparison of the $\text{Cu}^{2+}/\text{Cu}^+$ ratio between CuHHTP-5C (Supplementary Fig. 3) and CoHHTP-7C/CuHHTP-5C (Supplementary Fig. 12) showed that Cu^{2+} was reduced to Cu^+ after the Co_3HHTP_2 overlayer coating. This suggests that the electrical signal variation in the Co_3HHTP_2 layers induced by TMA adsorption can be transferred to the bottom Cu_3HHTP_2 layers, and changes the overall resistance of the film and enhancing its sensitivity.

Thus, this suggests that TMA gas exhibits stronger adsorption on Co_3HHTP_2 .

[R3-Q4] In Fig. 5e, the EtOH response under UV light (L1) is higher than under blue light. However, UV photons (3.13 eV) exceed the π - π^* transition energy (~ 2.9 eV). Does this cause ligand degradation? Include FTIR or Raman spectra of UV-exposed films to confirm structural stability.

[R3-A4] Thank you very much for the reviewer's valuable comments. As reviewer pointed out, we investigated repeatability test of the CuHHTP-15C sensor to 50 ppm EtOH under external UV light source. The results from 21 repeated exposures demonstrated that the CuHHTP-15C maintained a stable gas response to 50 ppm EtOH under UV illumination (Fig. R11). To further confirm structural stability, we conducted Raman spectroscopy measurements on the CuHHTP-15C film before and after UV exposure (Fig. R12). As the result, characteristics peaks of CuHHTP-15C did not exhibit any noticeable shift, indicating that no significant structural degradation occurred. Although UV LEDs were used in this study for EtOH sensing, blue LED illumination alone also showed sufficiently high performance. Moreover, because blue light has a longer wavelength (i.e., lower photon energy), it is less likely to cause ligand degradation in the cMOF structure. We have added Figs. R11 and R12 as Supplementary Figs. 17 and 18, respectively, and incorporated the corresponding explanations into the revised manuscript.

Fig. R11. Repeatability of the CuHHTP-15C sensor to 50 ppm of EtOH under external UV light source (response: 3600 s, recovery: 7200 s).

Fig. R12. a–b Raman spectroscopy (laser: 514 nm) results of CuHHTP-15C film of before (a) and after (b) 50 ppm EtOH repeatability test under UV light.

<After Correction> Results- Integration of cMOFs and μ LP (page 14, line 307)

(Original) For the UV μ LED, it was observed that prolonged use at high intensity caused damage to the conductivity of the MOF, so the experiment was conducted only at the L1 level. The detailed light-current-voltage (L-I-V) properties of the μ LED are summarized in **Supplementary Fig. 16**, and the actual emission of the μ LED is shown in **Fig. 5d**.

→ For the UV μ LED, since it was observed that prolonged use at high intensity can cause damage to the conductivity of the MOF, the experiment was conducted only at the L1 level. Nevertheless, even after prolonged UV L1 exposure for over 60 hours, the cMOF maintained its conductivity and EtOH sensing performance (**Supplementary Fig. 17**), and Raman spectroscopy confirmed that no significant ligand degradation occurred after the UV exposure (**Supplementary Fig. 18**). The detailed light-current-voltage (L-I-V) properties of the μ LED are summarized in **Supplementary Fig. 19**, and the actual emission of the μ LED is shown in **Fig. 5d**.

<After Correction> Supporting Information

Supplementary Fig. 17 | Repeatability of the CuHHTP-15C sensor to 50 ppm of EtOH under external UV light source (response: 3600 s, recovery: 7200 s).

[R3-Q5] On Page 15, Lines 315–318, The UV-vis spectra show a broad peak at 645 nm (ligand-to-metal charge transfer). How does this align with the μ LED wavelengths (395 nm, 455 nm)? Provide band diagrams to illustrate photoexcitation pathways.

[R3-A5] Thank you for your comments. The Tauc plot demonstrated that the energy of ligand-to-metal charge transfer (LMCT) is ~ 0.73 eV. Given that 395 nm and 455 nm correspond to photon energies of 3.14 eV and 2.73 eV, respectively, both wavelengths provide sufficient energy to excite the LMCT transition.

In this study, the large π - π^* transition energy gap (~ 2.9 eV) plays a dominant role in light activation, as the material is not fully activated without light illumination. In contrast, the LMCT-related low energy gap is known for the origin of room temperature conductivity, meaning it is already activated. But as we mentioned above, this small gap can also be activated by LED lights having enough energies.

To aid the reader's understanding, as suggested by Reviewer #3, a supplementary explanation is added to the caption of **Supplementary Fig. 23**.

[R3-Q6] In **Supplementary Fig. 24**, the temperature rise under μ LED illumination is $< 0.5^\circ\text{C}$. Were measurements taken under continuous operation (e.g., 1 hour)? Include long-term stability data to rule out cumulative heating effects.

[R3-A6] We appreciate the reviewer's comment. Temperature measurement of blue μ LED and blue μ LED +cMOF samples in **Supplementary Fig. 24** were conducted using an infrared micro-thermography measurement setup. The measurements were taken after illuminating the blue μ LED at L2 intensity for 5 minutes to reach saturation.

As the reviewer pointed out, long-term stability data is essential to rule out any cumulative heating effects. To address this concern, we operated both the blue μ LED-only sample and the Cu_3HHTP_2 -on-blue- μ LED sample under L2 illumination for one hour. Temperature was recorded at thirteen time points—every 5 minutes from the start to 60 minutes (**Figs. R13 and R14**). The results show no significant temperature increase in either sample, confirming that the cMOF activation is driven solely by μ LED illumination without notable photothermal effects. We have added **Figs. R13 and R14** as **Supplementary Fig. 25** and included a description in the manuscript.

While addressing the reviewer's comment, we realized that the original manuscript contained an unintentional error. We had stated that "*Temperature measurements were conducted under L2 light intensity for both the blue μ LED-only sample and the blue μ LED with the cMOF layer (Cu_3HHTP_2).*" However, the actual experiment (**Supplementary Fig. 24**) involved sweeping the μ LED forward bias from 1 V (LED off) to 4 V, and the L2 light intensity corresponds specifically to the condition at approximately 3.1 V. We have now corrected this discrepancy in the revised manuscript to accurately reflect the experimental conditions.

Fig. R13. Calibrated temperature profile of the blue μ LED-only sample under continuous L2 illumination.

Fig. R14. Calibrated temperature profile of the Cu_3HHTP_2 -on-blue- μ LED sample under continuous L2 illumination.

<After Correction> Results-Integration of cMOFs and μ LP (page 17, line 365)

(Original) To confirm that the sensing mechanism of the μ LED-integrated cMOF sensor is driven purely by photoactivation rather than photothermal-induced temperature increase, we employed an infrared micro-thermography system (**Supplementary Fig. 19**). Temperature measurements were conducted under L2 light intensity for both the blue μ LED-only sample and the blue μ LED with cMOF layer (Cu_3HHTP_2). The results showed a minimal temperature increase of less than $0.5\text{ }^\circ\text{C}$ in both samples.

→ To confirm that the sensing mechanism of the μ LED-integrated cMOF sensor is driven purely by photoactivation rather than photothermal-induced temperature increase, we employed an infrared micro-thermography system (**Supplementary Fig. 24**). ~~Temperature measurements were conducted under L2 light intensity for both the blue μ LED-only sample and the blue μ LED with cMOF layer (Cu_3HHTP_2).~~ Temperature measurements were conducted while operating the blue μ LED under a forward bias ranging from 1V (LED OFF) to 4V, for both the blue μ LED-only sample and the blue μ LED with the Cu_3HHTP_2 . The results showed a minimal temperature increase of less than $0.5\text{ }^\circ\text{C}$ in both samples. Under continuous blue μ LED (L2) illumination for 60 minutes, both samples exhibited minimal temperature changes of less than $0.5\text{ }^\circ\text{C}$ (**Supplementary Fig. 25**). This finding confirms that the gas sensing of the cMOF is facilitated by the activation of its energy gap through photoactivation by the μ LED, rather than by any temperature increase.

(a) Blue μ LED (L2)

(b) Blue μ LED (L2) with Cu_3HHTP_2

Supplementary Fig. 25 | a–b Calibrated temperature profiles under continuous L2 illumination. (a) Blue μ LED-only sample. (b) Cu_3HHTP_2 synthesized on blue μ LED sample.

[R3-Q7] In **Fig. 2h**, the NH_3 response of CuHHTP-1C (457.1%) is claimed superior to prior cMOF sensors. Compare response times (T_{90}) with literature to assess practicality for real-time applications.

[R3-A7] We appreciate the reviewer's insightful comment. As the reviewer pointed out, response time is a critical factor in assessing the practicality of gas sensors for real-world applications, in addition to response magnitude. We could easily compare responses of corresponding references, but it was hard to measure T_{90} of other literatures owing to their less-refined transient data.

Fig. R15. Relative responses of a Cu_3HITP_2 device to 0.5, 2, 5, and 10 ppm ammonia diluted with nitrogen gas.⁸ (Ref 8: *Angew. Chemie - Int. Ed.* 2015, 54, 4349–4352)

Fig. R16. 80 ppm NH_3 sensing transients for Cu_3HHTP_2 .⁴³ (Ref 43: *Chem. Mater.* 2016, 28, 15, 5264–5268)

For instance, in reference 8 (**Fig. R15**) and reference 43 (**Fig. R16**), it is quite difficult to determine the exact T_{90} point based on the provided figures.

Instead, we can compare the workplace exposure limits for NH_3 set by several agencies in the USA. OSHA has established a permissible exposure limit (PEL) of 50 ppm, averaged over an 8-hour work shift. Meanwhile, NIOSH and ACGIH recommend an exposure limit (REL) of 25 ppm for long-term exposure (averaged over 10 hours) and 35 ppm for short-term exposure (under 15 minutes). [New Jersey Department of Health, "Right to Know Hazardous Substance Fact Sheet: Ammonia (NH_3), <https://nj.gov/health/eoh/rtkweb/documents/fs/0084.pdf>"]

Our sensor easily surpasses these standards for practical safety applications, even when using our pristine Cu_3HHTP_2 sensor without machine learning processing. Please see **Fig. R17** below.

Fig. R17. Response transients of CuHHTP- x C ($x = 1, 3, 5, 7, 9, 11, 13,$ and 15) sensors to 5 ppm NH_3 .

NIOSH and ACGIH have established 35 ppm as the threshold for acute exposure over a 15 -minute period. In contrast, **Fig. R17** demonstrates that even at a much lower concentration of 5 ppm NH_3 , the CuHHTP- 1 C sensor achieves a detection rate of 26.4% after 15 minutes of exposure. Furthermore, even the thicker CuHHTP- 15 C sensor generates a response of 2.4% under the same conditions. This indicates that our sensor exhibits excellent performance as a standalone sensor.

Additionally, incorporating a CNN into the sensor system can further accelerate detection by several times. In this study, the raw sensor signal exhibited a T_{90} and T_{10} of several minutes. However, with CNN implementation, the types and concentrations of the four target gases were predicted within 1 min in most cases, and even when considering the 60 -second sliding time window, the overall prediction time remained within 2 minutes. This improvement is achieved because the CNN extracts key features from the transient signals generated by different gas types and concentrations, allowing the model to learn these patterns in advance. As a result, even in the early stage of the response—before full saturation—CNN can accurately predict gas type and concentration by analyzing the initial slope and signal profile during both the response and recovery phases. For a more detailed explanation of CNN, please refer to our response to **Reviewer #2 (R2-A3)**. Additionally, further details on CNN prediction time can be found in **Reviewer #3 (R3-A9)**.

To improve the readability of **Fig. 2h**, we added reference numbers to each corresponding point in the graph.

Fig.2 | LBL synthesis process, characterization, and gas testing results of Cu₃HHTP₂. h Comparison of CuHHTP-xC sensors in this study with previous cMOF-based gas sensors research (L: LBL-coated films, P: powders). (*Reference numbers are added.*)

[R3-Q8] On page 14, Lines 294, in **Fig. 5f**, NO₂ recovery rate improves under blue light (L2). Quantify the recovery rate for repeated cycles to validate long-term reversibility.

[R3-A8] Thank you for your comment. To assess the long-term reversibility of NO₂ sensing under blue light (L2), we conducted ten repeated cycles at 2 ppm NO₂. As shown in **Fig. R18**, the μ LED-induced photoactivation enables sufficient recovery and ensures stable long-term repeatability, confirming the reliability of our sensor system.

Fig. R18. Response transients of CuHHTP-5C sensors to 2 ppm NO₂ under blue μ LED illumination at L2 intensity over ten repeated cycles.

<After Correction> Results-Integration of cMOFs and μ LP (page 16, line 348)

(Original) The total power consumption of the four-sensor array was 587 μ W, keeping it below 1mW. With their optimal sensitivity, all sensors were reversible, enabling repeated and continuous gas detection with minimum power consumptions.

→ The total power consumption of the four-sensor array was 587 μ W, keeping it below 1mW. With their optimal sensitivity, all sensors were reversible, enabling repeated and continuous gas detection with minimum power consumptions. **Cycle tests under these optimized conditions (Supplementary Fig. 22) confirmed good repeatability, with μ LED photoactivation aiding recovery of the cMOF sensors.**

Supplementary Fig. 22 | a–d Repeatability test of optimized cMOF sensors under cyclic gas exposures. (a) CuHHTP-5C (Blue L2) sensor to 2 ppm NO₂ over 10 cycles. (b) CuHHTP-15C (UV L1) sensor to 50 ppm EtOH over 20 cycles. (c) CoHHTP-7C/CuHHTP-5C (Blue L2) sensor to 50 ppm TMA over 20 cycles. (d) CuHHTP-5C (Dark) sensor to 5 ppm NH₃ over 20 cycles.

[R3-Q9] On Page 18, Lines 364–365, the CNN predicts gas types within "tens of seconds." Specify the exact latency (e.g., 20 s vs. 60 s) and its dependence on sensor response kinetics.

[R3-A9] Thank you for your comment. In this study, we define the CNN prediction latency as the time between the onset of gas injection (indicated by the gray-shaded region in **Fig. 6b**) and the first correct prediction of the target gas. In most cases, accurate predictions were achieved from few seconds to one minute, demonstrating the practical speed of the algorithm.

As described in the CNN section of the manuscript, we applied a sliding time window approach in which gas response data are collected every second and accumulated into a 60-second window. Thus, an initial 60 seconds is required before the first prediction can be made. However, after this initial accumulation, predictions are generated in real time at 1-second intervals using updated sliding windows (e.g., 1–60 s, 2–61 s, and so on). Even when accounting for this initial buffering time, the overall latency remains under 2 minutes (comprising less than 1 minute of prediction latency plus the 60-second sliding time window), which is significantly shorter than the intrinsic response and recovery times of the cMOF sensors, typically around 20 minutes.

Regarding the reviewer's question about the dependence on sensor response kinetics, we note that the CNN algorithm performs pattern recognition based on the transient temporal profiles of gas responses. Therefore, the model can accurately classify gases even before the sensor reaches full equilibrium, as long as the early-stage response pattern contains sufficient distinguishing features. Although faster and more differentiable sensor responses can enhance the CNN's prediction performance, they are not strictly necessary. The algorithm is capable of extracting meaningful features from early transient behavior, allowing accurate predictions even in the presence of slow sensor kinetics.

Additionally, there exists a trade-off between prediction accuracy and latency depending on the choice of the sliding window size—shorter windows yield faster predictions with potentially reduced accuracy, whereas longer windows enhance prediction reliability at the cost of latency. For further details on this trade-off, we kindly refer the reviewer to our previous work (<https://pubs.acs.org/doi/10.1021/acsnano.2c09314>), especially **Supplementary Fig. 17** of that work.

<After Correction> Results-Deep learning-based cMOF e-nose system (page 19, line 410)

(Original) Additionally, while the response and recovery speed of the cMOF sensors themselves are somewhat slow, the CNN-based algorithm predicts gas types and concentrations within tens of seconds, making it highly practical.

→ Additionally, although the response and recovery times of the cMOF sensors are on the order of tens of minutes (**Supplementary Fig. 20**), the CNN-based algorithm, even when considering the 60-second sliding time window, can predict gas types and concentrations within two minutes, making it highly practical for real-time applications.

[R3-Q10] On Page 11, Lines 229–32, the Co/Cu alloy phase is hypothesized but not confirmed. Perform TEM/SAED or XRD to verify epitaxial growth and lattice matching.

[R3-A10] Thank you very much for your comment. Our hypothesis is based on two key points: It has already been proven that triphenylene-based ligands can form alloy structures when two different metal ions are mixed (*J. Am. Chem. Soc.*, 2020, 142 (28), 12367–12373). Additionally, it has been revealed that even when the two metals are mixed, the conductivity remains continuous through the ligand. Our Ni₃HHTP₂ and Co₃HHTP₂ overlayer coating process was conducted immediately and continuously after the formation of the Cu₃HHTP₂-based films, while the reaction was still incomplete. Therefore, we argued that a similar local alloy was formed, and based on these factors, we stated that “...are presumed to form a local alloy phase.”

However, there is no solid evidence to support these statements, as Ni₃HHTP₂ and Co₃HHTP₂ layers are too thin to characterize their crystallinity. We only confirmed that both layers exist on the surface of the Cu₃HHTP₂ films (**Supplementary Figs. 9 and 10**).

Supplementary Fig. 9 | **a** Cross-section view SEM image of NiHHTP-7C/CuHHTP-15C on Si/SiO₂ substrate. **b–e** Energy-dispersive X-ray spectroscopy (EDS) mapping of NiHHTP-7C/CuHHTP-15C on Si/SiO₂ substrate; **(b)** C element, **(c)** O element, **(d)** Ni element, and **(e)** Cu element. The substrate is tilted by 3° to observe the presence of both the Ni element in the overlayers and the Cu element in the base layer.

Supplementary Fig. 10 | **a** Cross-section view SEM image of CoHHTP-7C/CuHHTP-15C on Si/SiO₂ substrate. **b–e** Energy-dispersive X-ray spectroscopy (EDS) mapping of CoHHTP-7C/CuHHTP-15C on Si/SiO₂ substrate; **(b)** C element, **(c)** O element, **(d)** Co element, and **(e)** Cu element. The substrate is tilted by 3° to observe the presence of both the Co element in the overlayers and the Cu element in the base layer.

In this system, Cu₃HHTP₂ conducting films were already formed after 5 cycles. Any further coating has a minor effect on conductivity but rather contributes to the gas reaction through filtration or catalytic properties (**Fig. 2e**). Since our focus is on the catalytic properties of Ni₃HHTP₂ and Co₃HHTP₂, the crystallinity of these films was not a critical factor.

Therefore, we removed and revised the corresponding statement in the manuscript to avoid any potential controversy.

<After Correction> Results-Overlayer coating of $M_3\text{HHTP}_2$ ($M = \text{Ni}, \text{Co}$) on CuHHTP-5C films (page 12, line 248)

(Original) For methyl-oxygen interactions, the binding energies were relatively low across all MOFs, ranging from -0.33 eV to -0.37 eV, presumably due to the reduced influence of metal sites. Thus, this suggests that TMA gas exhibits stronger adsorption on Co_3HHTP_2 . The Co_3HHTP_2 overlayers are epitaxially stacked onto the Cu_3HHTP_2 base layers through continuous immersion cycles of ligand and metal solutions. During this process, the two 2D MOF layers are presumed to form local alloy phases, where Co and Cu coordinate through a single HHTP ligand in-plane, or the Co_3HHTP_2 and Cu_3HHTP_2 layers stack out-of-plane. As a result, redox reactions occurring at the Co_3HHTP_2 layer can influence the overall resistance of the film and, in the case of TMA, enhance sensitivity.

→ For methyl-oxygen interactions, the binding energies were relatively low across all MOFs, ranging from -0.33 eV to -0.37 eV, presumably due to the reduced influence of metal sites.

In addition, a comparison of the $\text{Cu}^{2+}/\text{Cu}^+$ ratio between CuHHTP-5C (Supplementary Fig. 3) and $\text{CoHHTP-7C}/\text{CuHHTP-5C}$ (Supplementary Fig. 12) showed that Cu^{2+} was reduced to Cu^+ after the Co_3HHTP_2 overlayer coating. This suggests that the electrical signal variation in the Co_3HHTP_2 layers induced by TMA adsorption can be transferred to the bottom Cu_3HHTP_2 layers, and changes the overall resistance of the film and enhancing its sensitivity.

~~Thus, this suggests that TMA gas exhibits stronger adsorption on Co_3HHTP_2 . The Co_3HHTP_2 overlayers are epitaxially stacked onto the Cu_3HHTP_2 base layers through continuous immersion cycles of ligand and metal solutions. During this process, the two 2D MOF layers are presumed to form local alloy phases, where Co and Cu coordinate through a single HHTP ligand in plane, or the Co_3HHTP_2 and Cu_3HHTP_2 layers stack out of plane. As a result, redox reactions occurring at the Co_3HHTP_2 layer can influence the overall resistance of the film and, in the case of TMA, enhance sensitivity.~~

[R3-Q11] In Fig. 6d, The MAE for concentration regression is 7.94%. Does this error vary with gas type (e.g., higher for TMA)? Include error distributions for individual gases.

[R3-A11] Thank you for your comment. The average concentration prediction error across the four target gases is indeed 7.94%, and as the reviewer pointed out, this error varies depending on the gas type. Since the CNN model predicts gas type and concentration based on the resistance change of cMOFs, a larger response generally leads to easier differentiation and more accurate predictions. Among the four gases, TMA at 50 ppm exhibited the highest error of 22.5%, which aligns with its relatively low response magnitude, making it more challenging for the model to distinguish. The error distributions for individual gases were originally shown in Supplementary Fig. 22, but have been moved to **Supplementary Table 2**, as the tabular format is more appropriate for this type of data.

Supplementary Table 2 | Prediction errors (mean absolute error; MAE) of the CNN model for gas types and concentrations.

Target gases							
EtOH		TMA		NH ₃		NO ₂	
50 ppm	11.1%	50 ppm	22.5%	10 ppm	10.0%	1 ppm	8.2%
100 ppm	8.1%	100 ppm	4.7%	20 ppm	6.1%	2 ppm	8.5%
200 ppm	2.7%	200 ppm	4.6%	50 ppm	3.2%	5 ppm	5.6%

[R3-Q12] In **Supplementary Fig. 18**, sensitivity decreases for NH₃ under light. Does this correlate with hole/electron recombination? Provide photoluminescence quenching or other data to elucidate carrier dynamics.

[R3-A12] Thank you very much for your comment. We realized that we had missed providing a detailed sequence of the photoactivation and recovery process. For NO₂ and NH₃, due to their strong adsorptive properties (*Adv. Sci.*, 2019, 6(21), 1900250. *Angew. Chem. Int. Ed.*, 2019, 131(42), 15057-15061) the forward reactions in Equations 1 and 2 proceed favorably.

In this case, if the number of electrons or holes increases, the reverse reaction can be enhanced, thereby accelerating the desorption of the adsorbed NH₃ and NO₂ gases.

In this study, we utilized blue μ LED (2.73 eV) or UV μ LED (3.14 eV), which provide energy to excite the π - π^* transition energy gap (\sim 2.90 eV) of the HHTP-based MOF, generating photoelectrons and thereby facilitating the recovery of adsorbed gases. In contrast, NH₃ sensing is sufficiently reversible even under dark conditions, and therefore, it is not necessary to enhance the recovery reaction through light activation at the expense of sensitivity. As such, we included the CuHHTP-5C sensor under dark conditions as a NH₃ sensors in the sensor array (**Fig. 6**).

We have identified the lack of detailed explanation in the main text and have now supplemented it with the following additional description.

<After Correction> Results-Integration of cMOFs and μ LP (page 15, line 320)

(Original) This is presumably because the high-energy UV light not only increases the reaction sites for the neutral methyl group but also accelerates the desorption of basic amine sites of TMA gases. In contrast, when sensing basic NH₃ and acidic NO₂ using CuHHTP-5C sensors, the reverse reaction became more dominant under light activation due to their excessive adsorption, even in dark conditions as shown in Eqs. (1) and (2) (**Supplementary Fig. 17**).

Although the correlation between light intensity and the response is not clearly demonstrated, the sensitivity decreases when the sensor is exposed to light. For NO₂, light activation is beneficial due to its strong adsorption characteristics, which significantly improve the recovery rate, thereby enhancing reversibility (**Fig. 5f**)

→ This is presumably because the high-energy UV light not only increases the reaction sites for the neutral methyl group but also accelerates the desorption of basic amine sites of TMA gases. **In contrast, when sensing basic NH₃ and acidic NO₂ using CuHHTP-5C sensors, the forward reaction is strongly favored, as shown in Eqs. (1) and (2) (Supplementary Fig. 20). In particular, NO₂⁻ adsorbates are irreversibly bound to the MOF surfaces. At this stage, light illumination can generate electron-hole pairs, thereby accelerating the reverse reaction and promoting the desorption of adsorbates.**

Although hindrance of the forward reaction can limit gas sensitivity, light activation is beneficial in the case of NO_2 , significantly improving the recovery rate and thereby enhancing reversibility (**Fig. 5f**).

[R3-Q13] In Fig. 4, DFT calculations focus on TMA binding. Extend simulations to NH₃ and NO₂ to explain selectivity trends in CuHHTP-*x*C sensors. Meanwhile, please further supplement the DFT calculation details, such as energy cutoff (ENCUT) and other parameter settings.

[R3-A13] Thank you for your insightful comment. We conducted a comparison of TMA adsorption energies on different M₃HHTP₂ frameworks because the sensing responses to TMA were notably enhanced upon coating with Co₃HHTP₂ layers. In contrast, the responses to other gases did not show meaningful improvement depending on the presence of Co₃HHTP₂ or Ni₃HHTP₂ layers. Therefore, we believe it is not necessary to perform separate DFT calculations for each type of conductive MOF layer.

As we discussed, the sensing response can be influenced not only by catalytic effects but also by the thickness of the films. In the cases of NH₃ and NO₂, the responses are highly dependent on film thickness, and the additional coating with Co₃HHTP₂ or Ni₃HHTP₂ layers inevitably increases the thickness, which may reduce the sensitivity—especially from the perspective that thinner films generally exhibit higher sensitivity.

However, as shown in Fig. 2d, TMA sensing response does not significantly vary with film thickness, indicating that thickness is not a dominant factor in this case. Interestingly, after coating with Co₃HHTP₂, the TMA response increased significantly. This allows us to isolate the catalytic effect from the thickness effect, thereby justifying the DFT calculations specifically for TMA.

In response to Reviewer #3's question, we will include this clarification in the revised manuscript. Additionally, the supplementary data related to the DFT calculations have been added to the Methods section for completeness.

<After Correction> Results-Overlayer coating of M₃HHTP₂ (M = Ni, Co) on CuHHTP-5C films (page 12, line 259)

(Original) As a result, redox reactions occurring at the Co₃HHTP₂ layer can influence the overall resistance of the film and, in the case of TMA, enhance sensitivity.

→ ~~As a result, redox reactions occurring at the Co₃HHTP₂ layer can influence the overall resistance of the film and, in the case of TMA, enhance sensitivity.~~ With regard to EtOH, NH₃, and NO₂ gases, DFT calculations were not conducted because their responses did not show noticeable improvement upon overlayer coating. Furthermore, unlike TMA, which exhibits consistent responses regardless of film thickness (Fig. 2d), the responses to EtOH, NH₃, and NO₂ are strongly influenced by the thickness of the films. Therefore, it is challenging to decouple the effects of film thickness from the catalytic effects of the overlayers in these cases.

<After Correction> Methods-Density functional theory (DFT) calculations (page 23, line 499)

(Original) DFT calculations were carried out using the Vienna Ab Initio Simulation Package (VASP) version 5.4.1⁴³. The monolayer structure of M₃HHTP₂ (M = Co, Ni, Cu) was modeled with a 20 Å vacuum slab, which was adapted from prior research⁴⁴. The calculations employed the GGA-PBE functional using the PAW method. To account for van der Waals

interactions between the TMA molecule and the MOF layer, Grimme's D3 dispersion correction was included^{45,46}. Convergence criteria for electronic and ionic optimizations were set at 10^{-5} eV and 0.003 eV/Å, respectively. Gaussian smearing was applied with a broadening factor of 0.05 eV. The k-points sampling was performed using a gamma-centered $1 \times 1 \times 1$ grid within the Brillouin zone. The binding energy of the TMA molecule with the MOF was calculated using the formula $E_{\text{binding}} = E_{\text{MOF+TMA}} - E_{\text{MOF}} - E_{\text{TMA}}$.

→ DFT calculations were carried out using the Vienna Ab Initio Simulation Package (VASP) version 5.4.1⁴³. The monolayer structure of $M_3\text{HHTP}_2$ ($M = \text{Co, Ni, Cu}$) was modeled with a 20 Å vacuum slab, which was adapted from prior research⁵⁰. The Perdew–Burke–Ernzerhof (PBE) functional within the generalized gradient approximation (GGA) was used in conjunction with the projector augmented wave (PAW) method. A plane-wave energy cutoff (ENCUT) of 520 eV was applied, which is $1.3 \times$ higher than the default ENMAX of the pseudopotentials, and the precision level was set to Accurate (PREC). Spin-polarized calculations (ISPIN = 2) were employed to capture magnetic effects, and a Gaussian smearing method (ISMEAR = 0) with a width of 0.05 eV (SIGMA) was used for electronic occupations. To account for van der Waals interactions between the TMA molecule and the MOF layer, Grimme's D3 dispersion correction with zero damping (IVDW = 11) was included^{51,52}. Convergence criteria for electronic and ionic optimizations were set at 10^{-5} eV (EDIFF) and 0.003 eV/Å (EDIFFG), respectively. The optimizations were performed using the conjugate gradient algorithm (IBRION = 2), with no symmetry constraints imposed (ISYM = 0). The k-points sampling was performed using a gamma-centered $1 \times 1 \times 1$ grid within the Brillouin zone. The binding energy of the TMA molecule with the MOF was calculated using the formula $E_{\text{binding}} = E_{\text{MOF+TMA}} - E_{\text{MOF}} - E_{\text{TMA}}$.

[R3-Q14] On Page 14, Lines 278–282, EtOH responses increase with light intensity. Is this due to photoactivated surface reactions or bulk conductivity changes? Perform impedance spectroscopy or other methods under illumination for further analysis.

[R3-A14] Thank you for your valuable comments. Unlike strongly adsorptive gases, the amount of EtOH that reacts with MOFs is relatively small. This is because EtOH does not significantly extract or inject charge carriers into the MOF framework; instead, it interacts weakly through their polarity, resulting in relatively weak binding to the MOF surface (*Angew. Chem. Int. Ed.*, 2021, 133(34), 18814-18820). When the reactivity is low, it becomes crucial to increase the number of available reaction sites or the reaction probability. Upon light irradiation, additional electron–hole carriers are generated within the MOF, substantially enhancing the probability of reaction with EtOH.

If the light-induced sensitivity enhancement were primarily due to improved bulk conductivity of the MOF, similar enhancements would be expected for all gases. However, in our experiments, a significant enhancement was observed only for EtOH, while other gases showed decreased responses. This suggests that the enhancement is not merely due to a change in the intrinsic physical properties of the MOF, but rather due to modulation of the interaction between the MOF and the EtOH molecules.

We are currently planning follow-up studies to further investigate this surface photoactivation mechanism, based on computational approaches such as DFT, as well as a variety of optical characterization techniques.

[R3-Q15] On Page 17, Lines 342–343, data augmentation applies $\pm 10\%$ noise. Does this artificially inflate accuracy? Compare model performance with/without augmented data.

[R3-A15] We appreciate the reviewer’s comment. The $\pm 10\%$ noise was introduced for data augmentation to develop a deep learning model with a more robust and diverse dataset. The choice of 10% noise was based on **Supplementary Fig. 5**, where three repeated experiments using different sensors showed an approximate 10% deviation, making this a reasonable threshold. If cMOFs had been synthesized using a different method, such as powder-based fabrication instead of the LBL method, the sensor-to-sensor deviation would have been significantly higher. Naturally, reducing this deviation would enable the development of a more accurate e-nose model. Additionally, to further investigate the impact of data augmentation, we trained the model with data augmented using 30% noise, and the results are presented in **Fig. R19**. We trained the CNN model with 30% deviation while keeping all other hyperparameters, model structure, and 2900 epochs the same as in the original training. The classification accuracy remained high at 98.8%, but the regression error significantly increased to 49.6% (mean absolute error; MAE), compared to 7.94% in the original model. These results highlight the crucial role of sensor-to-sensor reproducibility and uniformity in achieving reliable and accurate gas concentration predictions.

Figure R19. Comparison of CNN model performance with data augmentation using $\pm 30\%$ deviation. (a) Real-time prediction results for four different gases: EtOH (E), TMA (T), NH₃ (A), and NO₂ (N). (b) Confusion matrix showing classification results. (c) Regression results for concentration prediction (normalized from 0 to 1).

[R3-Q16] On Page 9, Lines 172–174, CuHHTP-1C and CuHHTP-3C show high NH₃ responses but are excluded due to deviations. Could a hybrid array (thick + thin films) improve overall performance?

[R3-A16] We appreciate the reviewer's insightful comment. In **Fig. 2h**, while CuHHTP-1C and CuHHTP-3C exhibit high sensitivity to NH₃, their thin-film nature results in pronounced sensor-to-sensor variation, which is why CuHHTP-5C was chosen for this study (please see **Supplementary Fig. 5**). However, we also confirmed that the deviation for CuHHTP-3C remains within an acceptable range.

Incorporating a hybrid array consisting of both thin and thick films for NH₃ sensing could leverage the advantages of each—high sensitivity from thin films and high uniformity from thick films. By integrating this approach with deep learning, such a system could enhance overall accuracy in gas prediction, making it a promising strategy for future high-precision gas sensing applications.

As shown in **Fig. 6a**, the e-nose system proposed in this study is already composed of multiple sensors with varying cMOF thicknesses, as suggested by the reviewer. Some sensors employ thin films, while others, such as CuHHTP-15C (with multiple LBL cycles) or CoHHTP-7C/CuHHTP-5C (with overlayer structures), are relatively thicker. This indicates that the reviewer's suggestion is fully consistent with the design concept implemented in our system.

Again, we thank you very much for your valuable comments and suggestions. Your favorable decision will be greatly appreciated.

Sincerely,

Inkyu Park, Ph.D.

Professor

Department of Mechanical Engineering & KI for the NanoCentury
Korea Advanced Institute of Science and Technology (KAIST)

Reviewer #1 (Remarks to the Author):

The authors have addressed most of my questions. But there is still something unclear.

Thank you once again for your comment. We have addressed your previous comments point by point and revised the manuscript accordingly. However, we noticed that one part still remained unclear, and we have further clarified and reinforced that part in this revision. Please see our detailed responses below.

[R1-Q1] From R1-Q3 and R1-Q4 in revision 1: Could you please explain why the roughness of the cMOF film does not significantly affect the gas sensing performance. It is risky to have such statements.

[R1-A1] Thank you for your comment. In our first revision, we noted that the films synthesized via the LBL method exhibited agglomerated particles on their surfaces. However, EDS analysis confirmed that these agglomerates are simply composed of $\text{Cu}_3(\text{HHTP})_2$ (**Supplementary Fig. 6**), as no impurities were detected other than C, Cu, O, and Si.

Both sensor platforms used in this study—the bare Si substrate and the actual micro-LED (μLED) gas sensor—have identically patterned interdigitated electrodes (IDEs) of $100 \times 100 \mu\text{m}^2$ with $5 \mu\text{m}$ spacing (**Fig. R1** and **Supplementary Fig. 1**). The sensitivity will be critically affected if the agglomerated particles are too large to cover the bottom conduction path, but the particle size was small enough not to hinder gas diffusion. As shown in **Fig. R2**, **Supplementary Figs. 4** and **6**, the lateral scale of the surface roughness and particle distribution is much smaller than the size and spacing of the IDEs. This μLED -based chemiresistor operates by measuring the resistance between the IDE fingers and the agglomerated particles are randomly distributed within the gaps between these electrodes.

In addition, based on the SEM images (**Supplementary Fig. 4**), the agglomerated particles are not considered part of the continuous sensing layer that determines the overall conductivity, but rather appear as loosely deposited particles on the top surface. Therefore, as mentioned earlier, if they do not hinder gas diffusion, they are unlikely to have any significant impact on gas sensitivity and may only contribute to minor sample-to-sample variations.

Therefore, while surface roughness may cause greater sensor-to-sensor variation in thinner films, its impact becomes negligible as the film gets thicker (**Supplementary Fig. 5**). Although the surface may exhibit some roughness in the lateral directions, this does not significantly affect the vertical thickness buildup trend during LBL cycles (**Supplementary Fig. 4**). As

repeatedly emphasized throughout the manuscript, it is ultimately the film thickness of the cMOF that plays a key role in determining the gas sensing performance.

Finally, the presence of agglomerated particles is mainly due to the rapid LBL process employed in this work (immersing the substrate in the ligand solution for 2 minutes, followed by 1 minute in the metal solution, with brief cleaning steps). Extending the reaction and cleaning times could further improve surface uniformity by removing residual salts more effectively at each step. However, from a production perspective, we adopted the shorter cycle times because the current sensor already demonstrates excellent performance and uniformity (with less than 10% variation). Nevertheless, we plan to further optimize the reaction and cleaning steps in future work to enhance the film quality.

Fig. R1. Overview of the micro-LED platform and patterned IDEs used for gas sensing. (a, b) correspond to Fig. 1, and (c, d) to Fig. 5 in the manuscript, respectively.

Supplementary Fig. 1 | Au-interdigitated electrodes (IDEs) on bare Si substrates and a home-made gas sensing chamber that simultaneously measures multiple gas sensors.

Fig. R2. AFM image of CuHHTP-5C on Si substrate.

Supplementary Fig. 4 | a–h SEM images of CuHHTP- x C at different coating cycles ($x= 1—15$): (a) 1C, (b) 3C, (c) 5C, (d) 7C, (e) 9C, (f) 11C, (g) 13C, and (h) 15C. The CuHHTP-1C films were too thin to be measured accurately, and the CuHHTP-3C films exhibited discontinuities in the cross-sectional view. (The thickness of thin film was measured 10 times to calculate average.)

Supplementary Fig. 5 | a–d Gas sensing transients of CuHHTP-*x*C sensors (*x* = 1, 3, 5, 7, 9, 11, 13, and 15) to (a) 100 ppm ethanol (EtOH), (b) 100 ppm trimethylamine (TMA), (c) 5 ppm ammonia (NH₃), and (d) 5 ppm nitrogen dioxide (NO₂), under dark conditions. All results represent the average values from multiple different sensors (gray shading: standard deviation).

Supplementary Fig. 6 | **a–c** Top-view SEM image and EDS analysis of CuHHTP-1C, CuHHTP-3C, and CuHHTP-5C. **(i)** Top view SEM image of CuHHTP-5C on Si substrate. **(ii–v)** EDS mapping of CuHHTP-5C on Si substrate; **(ii)** C element, **(iii)** Cu element, **(iv)** O element, and **(v)** Si element. The particles on the film surfaces are confirmed as the agglomerated Cu_3HHTP_2 particles. These particles are grown through the rapid LBL process to optimize the engineering process, but this does not critically affect the gas sensing properties, as the batch-to-batch deviations remain below 10% (see Supplementary Table 1).

Reviewer #2 (Remarks to the Author):

All the issues I concerned have been well addressed. Now it could be accepted for publication.

We sincerely thank Reviewer #2 for the insightful comments. Your feedback helped us structure the manuscript more clearly and present our claims with greater scientific and logical support. We believe the revisions have strengthened the overall quality of the paper.

Reviewer #3 (Remarks to the Author):

Accept as is.

We are sincerely grateful to Reviewer #3 for the thorough and multifaceted review. Your careful and detailed questions prompted us to critically examine various aspects of our work, which greatly contributed to enhancing the completeness and rigor of the manuscript. Thanks to your thoughtful input, the study has become more robust and well-rounded.